# Spatial variance of spring phenology in temperate deciduous forests is constrained by background climatic conditions

Marc Peaucelle [1,2,6*], Ivan A. Janssens [3], Benjamin D. Stocker [1,2,7], Adrià Descals Ferrando [1,2], Yongshuo H. Fu[4], Roberto Molowny-Horas[1], Philippe Ciais [5] & Josep Peñuelas [1,2]

Leaf unfolding in temperate forests is driven by spring temperature, but little is known about the spatial variance of that temperature dependency. Here we use in situ leaf unfolding observations for eight deciduous tree species to show that the two factors that control chilling (number of cold days) and heat requirement (growing degree days at leaf unfolding, $GDD_{req}$) only explain 30% of the spatial variance of leaf unfolding. Radiation and aridity differences among sites together explain 10% of the spatial variance of leaf unfolding date, and 40% of the variation in $GDD_{req}$. Radiation intensity is positively correlated with $GDD_{req}$ and aridity is negatively correlated with $GDD_{req}$ spatial variance. These results suggest that leaf unfolding of temperate deciduous trees is adapted to local mean climate, including water and light availability, through altered sensitivity to spring temperature. Such adaptation of heat requirement to background climate would imply that models using constant temperature response are inherently inaccurate at local scale.

[1] CREAF, Cerdanyola del Vallès, Barcelona 08193 CA, Spain. [2] CSIC, Global Ecology Unit CREAF-CSIC-UAB, Bellaterra, Barcelona 08193 CA, Spain. [3] Department of Biology, University of Antwerp, Wilrijk 2610, Belgium. [4] College of Water Sciences, Beijing Normal University Beijing, Beijing, China. [5] Laboratoire des Sciences du Climat et de l'Environnement, UMR 1572 CEA-CNRS UVSQ, 91191 Gif sur Yvette, France. [6]Present address: Computational and Applied Vegetation Ecology - CAVElab, Department of Environment, Faculty of Bioscience Engineering, Ghent University, Gent, Belgium. [7]Present address: Institute of Agricultural Sciences, Department for Environmental Systems Science, ETH Zürich, Universitätsstrasse 2, 8006 Zürich, Switzerland. *email: mpeau.pro@gmail.com

The temporal variation in spring leaf unfolding (LU), including the effect of climatic warming, has been extensively documented[1–6]. In temperate and boreal regions, temporal variation of LU is dominated by the occurrence of warm spring temperatures[2], typically quantified by growing degree days[7–9] (GDD), with LU occurring when the supplied GDD equals the required GDD. The GDD requirement of LU (hereafter $GDD_{req}$) is commonly defined as the accumulated air temperature above a temperature threshold over the preseason. $GDD_{req}$ varies from year to year in response to differences in the occurrence of low winter temperatures[9,10] (chilling, hereafter NCD), and constraints imposed by optimal daylength[11]. Also, the availability of water and light have been shown to affect LU and its GDD requirement[11–13].

While the temporal variation in LU and its $GDD_{req}$ is extensively studied, the spatial heterogeneities of LU, and especially of its controls, have been much less studied. Land surface models assume that the drivers of the temporal variation of LU are also determining the spatial gradients in LU, but to our knowledge, this assumption has not been thoroughly tested. The temperature response of LU is often considered constant (spatially and temporally uniform), albeit species-dependent, in phenology models[14]. However, when applied at the regional scale, LU models were not able to accurately reproduce the observed spatial variation of LU[14–17].

Recent warming trends have advanced LU for virtually all temperate-zone tree species, but there is large spatial heterogeneity in these advancing LU trends at the global scale[18,19]. At the regional scale, different warming trends among sites have been proposed as the main cause for these spatially differing trends in mean LU date between high and mid-latitudes[20,21], low and high elevations[22–24] or coastal and inland areas[25,26]. Precipitation has also been identified as a key controlling factor of spatial differences of vegetation green-up in arid and semi-arid regions[12,27], while at high latitudes, photoperiodic controls were proposed as an evolutionary safety mechanism that mitigates the risk of frost damage[28,29] and causes LU to occur later at higher latitudes.

Studies assessing the effects of multiple environmental variables have highlighted complex and species-specific behaviors[12,27], mainly due to the mixing of temporal and spatial aspects of phenology, rather than trying to deconvolute these two components. Indeed, the spatial and temporal variances of phenology are expected to reflect two different mechanisms of control. First, short-term, fast responses to changing weather should drive temporal variations in LU and its $GDD_{req}$, aiming to maximize fitness under inter-annually varying weather conditions. Second, an adaptive response to local biogeographical conditions may maximize tree fitness under the local long-term mean climatic conditions and would select for spatially optimized LU and climate sensitivity, inducing spatial variation therein. Biogeographic constraints on LU include all environmental variables that impose long-term adaptation of LU and its $GDD_{req}$ to optimize fitness under local conditions. These include climatic variables, such as site-specific occurrence of late frost events, drought occurrence, low or high light extremes, that may need to be avoided and therefore require shifts in growing season to enable maximum tree fitness. Also, site-specific interactions with neighboring competitors, pathogens and herbivores may induce spatial differences in LU and its weather dependency, in order to maximize tree fitness. Taken together, this suggests a complex response of plant phenology to climate change, but also that models of LU that apply spatially uniform parameters may not capture the observed patterns of LU and its GDD requirement.

The key hypothesis that we explore in this study is that long-term mean background biogeographical conditions determine the spatial heterogeneity of spring LU and its $GDD_{req}$, reflecting evolutionary mechanisms through which plants have adjusted their growth strategies in order to maximize their fitness under those specific biogeographical conditions. We argue that biogeographic constraints on plant phenology can be detected by analyzing the spatial response of stands long-term mean LU and $GDD_{req}$, instead of their inter-annual variability. In this respect, we hypothesize that not only spring, but also mean growing season conditions are important controls of the spatial differences in leaf phenology among different locations. If so, these evolutionary mechanisms control the sensitivity of LU to short-term spring temperature variations, and consequently are key components of the observed spatial variability in LU and its $GDD_{req}$.

Here, we relate spatial variations in LU and $GDD_{req}$ in 8 dominant European deciduous tree species (see Methods) to long-term cross-sites differences in temperature, radiation and aridity (Supplementary Table 1). We combine long-term LU observations at 27790 sites over 1970–2016 (Supplementary Fig. 1) with climatic data from CRU-NCEP[30] at a spatial resolution of 0.5° (Supplementary Figs 2 and 3) to estimate the effects of long-term mean temperature, light (radiation and daylength) and aridity as determinants of the spatial variance of LU and $GDD_{req}$ using statistical models. Our study provides evidence for a significant control of leaf unfolding by long-term background climatic conditions across sites, potentially representing long-term adaptation of species. For all species: (1) the spatial variance of LU and $GDD_{req}$ supports previous correlations with temperature and chilling requirement, but is also determined by radiation intensity (W m$^{-2}$) and site aridity, (2) the spatial variance of $GDD_{req}$ is better explained by preseason radiation intensity than day length at LU, (3) LU and $GDD_{req}$ are more sensitive to water availability on drought-prone sites and (4) LU and $GDD_{req}$ respond both to long-term preseason and growing-season climatic conditions. These findings suggest that at least two mechanisms influence spring phenology: (i) the direct sensing of meteorological conditions during spring to optimize the restart of plant activity and (ii) the long-term adjustment of bud sensitivity to spring meteorological conditions in order to cope with growing season pressures at sites. It also suggests that common $GDD_{req}$ and NCD metrics used for simulating the temporal variability of LU are not suitable for spatial studies and should be used with caution.

## Results & Discussion

**Importance of background climate for LU spatial variability.** Among-site spatial variability of long-term mean annual LU (day of year -DoY-) and $GDD_{req}$ (°C; see Methods) was of the same order of magnitude than the typical interannual variation in LU and $GDD_{req}$ (Table 1). Figure 1 and 2 present the importance of each climatic variable (taking into account co-linearity and spatial autocorrelation; see Methods) in determining median LU date and $GDD_{req}$. The full model, including long-term temperature, incident surface radiation and water availability predictors, explained 61 ± 7% of the spatial variance of LU among species (regression coefficients in Supplementary Table 2). Chilling and $GDD_{req}$, which are functions of temperature and are additionally modulated by possible biotic adaptation to site-specific conditions, accounted for only half of the spatial variance of LU (Fig. 2a). Average preseason temperature (TP) was positively correlated with site LU and was selected as the best predictor instead of $GDD_{req}$ for half of the studied species. In addition to preseason temperature conditions, growing season temperature (TG) was found to explain 29 ± 5% of the spatial variability in LU. Considering all temperature-related variables as predictors, including preseason and growing season, models captured 52 ± 6% of total LU's spatial variance.

**Table 1 Observed LU and GDD$_{req}$ variability.**

| Species | Intra-site LU sd | Inter-Site LU sd | Intra-site GDD$_{req}$ sd | Inter-site GDD$_{req}$ sd | # sites | # drought prone sites |
|---|---|---|---|---|---|---|
| A. hippocastanum | 8.9 | 8.4 | 91.0 | 80.0 | 4429 | 1052 |
| A. glutinosa | 9.7 | 8.7 | 105.4 | 90.2 | 3207 | 687 |
| B. pendula | 8.95 | 7.5 | 85.5 | 78.0 | 4577 | 1079 |
| F. sylvatica | 7.3 | 6.5 | 105.0 | 100.0 | 3974 | 785 |
| F. excelsior | 8.2 | 7.5 | 126.8 | 106.9 | 3003 | 518 |
| Q. robur | 7.7 | 7.9 | 110.9 | 96.5 | 3995 | 892 |
| S. aucuparia | 7.7 | 7.5 | 93.4 | 132.8 | 2327 | 472 |
| T. cordata | 8.1 | 7.6 | 97.2 | 81.4 | 2278 | 262 |

Description of the intra and inter-site variability in observed leaf unfolding (LU) dates from the PEP dataset and the corresponding estimated Growing Degree Days requirement (GDD$_{req}$), as well as the number of sites (#) per species. Source data are provided as a Source Data file

Long-term mean LU across sites was also significantly positively correlated with long-term growing season mean incident shortwave radiation (SWG, visible and near infrared) for all species (Fig. 2a), while no clear effect of long wave radiation (LWG, infrared) was observed. SWG contributed an additional $9 \pm 2\%$ of explained variance in LU. The importance of SWG in the model determining LU was low compared to that of temperature (Fig. 1a), but still statistically significant (Supplementary Table 2). We also examined the relations between LU and precipitation (P), soil-moisture content (SM) and the ratio of actual to potential evapotranspiration[31] ($\alpha$E; see Methods). Long-term mean LU was significantly and negatively correlated to long-term mean growing-season P (PG) for A. hippocastanum, B. pendula, F. sylvatica, F. excelsior and Q. robur, for which it captured $4.0 \pm 0.6\%$ of LU's variance. Only T. cordata exhibited a negative correlation with preseason P (PP) and SM (SMP). Growing season SM (SMG) was not correlated to LU, while $\alpha$E showed a weak, but significant, positive correlation for F. sylvatica and F. excelsior, consistent with the relationship with PG (Fig. 2a; Supplementary Table 2).

**Importance of background climate for GDD$_{req}$ spatial variability.** Long-term temperature, incident surface radiation and water availability explained $54 \pm 12\%$ of the spatial variance of GDD$_{req}$ among species (Figs 1b and 2b; regression coefficients can be found in Supplementary Table 3). Large differences in the impact of long-term conditions on GDD$_{req}$ are observed among species, with background climate explaining between 33% of the spatial variance in GDD$_{req}$ for B. pendula and 73% for S. aucuparia.

For all species, GDD$_{req}$ was negatively correlated with NCD and positively correlated with TG, together explaining $52 \pm 4\%$ of the spatial variance in GDD$_{req}$ (Fig. 2c). The proportion of spatial variance in GDD$_{req}$ explained by preseason incoming shortwave radiation (SWP) was about 30%. Day length was selected as a predictor of GDD$_{req}$ only for A. hippocastanum and S. aucuparia. Growing season light conditions played a minor role, albeit still statistically significant, with incoming longwave radiation (LWG) capturing about $4.0 \pm 1.5\%$ of GDD$_{req}$'s explained variance (Fig. 2c). GDD$_{req}$ was significantly correlated with both PG and PP, capturing together around $7.5 \pm 2.3\%$ of GDD$_{req}$ variance. However, no significant correlations were found with soil moisture, nor $\alpha$E.

**Different response of phenology in drought-prone sites.** While precipitation was a weaker determinant of the spatial variance in GDD$_{req}$ than temperature and light, for all species we observed that the spatial variance in GDD$_{req}$ decreased with increasing drought stress ($\alpha$E) during the growing season (Fig. 3; Supplementary Fig. 4), suggesting that frequent and/or more intense water stress during the growing-season results in adaptation of

spring phenology in the long-term. Based on the observed relationships in Fig. 3, we therefore selected drought-prone sites ($\alpha_E < 0.9$) to assess if their responses to temperature, radiation and water differed (results for wet sites can be found in Supplementary Fig. 5 and Supplementary Tables 6 and 7).

At drought-prone sites, TP outcompeted GDD$_{req}$ as the main determinant of the spatial variance of LU. The contribution of temperature (NCD, GDD$_{req}$, TP and TG) did not change at drought-prone sites relative to all sites (Fig. 2b). However, the relative importance of preseason precipitation (PP) was doubled at those drought-prone sites compared to the wet-sites analysis (Supplementary Tables 4 and 6). Across the drought-prone sites, LU was negatively correlated with PP for A. hippocastanum, A. glutinosa, B.pendula, F. sylvatica and Q. robur. Instead, for A. glutinosa and S. aucuparia, LU was significantly and positively correlated with PG. LU also was positively correlated with $\alpha$E for F. excelsior, suggesting a differential effect of summer water availability on LU for these three species when grown at drought-prone sites. We also observed a smaller and divergent effect of incoming radiation on LU at drought-prone sites (Fig. 2b) compared to the wet-sites analysis (Supplementary Fig. 5).

As for LU, the relative importance of temperature and incoming radiation as determinants of the spatial variance of GDD$_{req}$ across drought-prone sites did not differ from the other sites (Fig. 2d). However, we observed a significant contribution of water availability at drought-prone sites compared to other sites. GDD$_{req}$ was negatively correlated with PG and positively correlated with PP, for all species except for T. cordata. GDD$_{req}$ was also negatively correlated with $\alpha$E for A.hippocastanum and A.glutinosa, while being positively correlated for T. cordata, which was consistent with the observed correlation with PG (Fig. 2e, Supplementary Tables 5 and 7).

While we observed significant differences in explaining LU and GDD$_{req}$ spatial variance at drought-prone sites, the same approach showed no differences between warm/cold sites or low/high light sites.

**GDD and NCD did not fully capture LU spatial variance.** We focused on the spatial variance of spring LU and its GDD$_{req}$ requirement. If the determinants of the temporal variation of LU and GDD$_{req}$ would not fully explain their spatial variation, this would imply adaptation of spring phenology to long-term local background environmental conditions. Results revealed that temperature and the interplay between chilling and heating during winter and spring, the main determinants of the temporal variation of LU, also were important factors controlling the spatial variation of spring LU, in line with previous studies[9,10]. However, we also showed that these chilling and heat requirement metrics only captured 30% of the total spatial variance in LU for all species (Fig. 2a, Supplementary Table 2). The positive correlation observed between LU and TP was not expected.

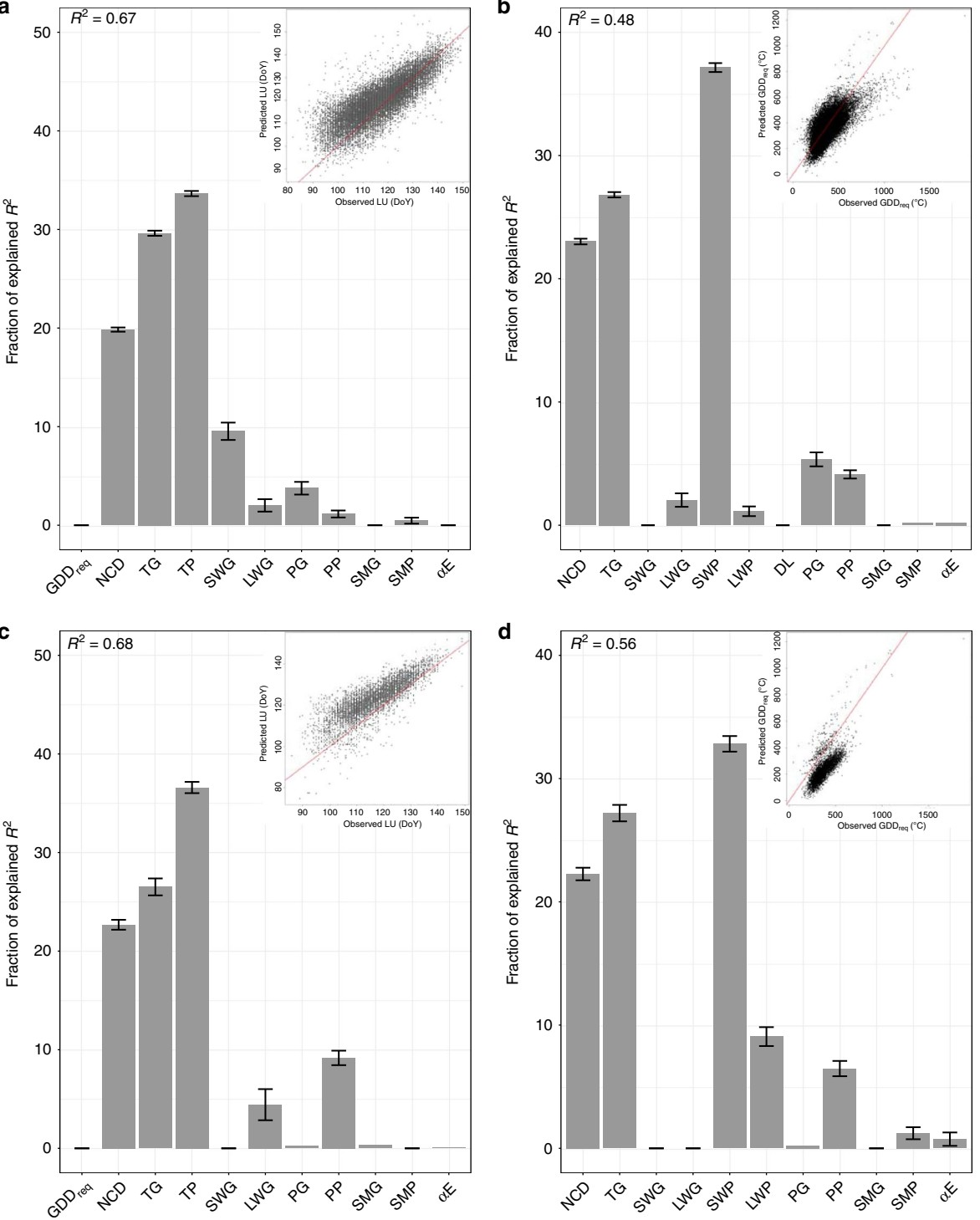

**Fig. 1** Effect of background climate on LU and GDD$_{req}$. Relative importance (%) of each variable in estimating **(a)** observed LU date for all sites and species (n = 27790), **(b)** GDD$_{req}$ for all sites and species, **(c)** LU date for drought-prone sites (n = 5747) and **(d)** GDD$_{req}$ for drought-prone sites. See the last row of Fig. 2a–d (ALL) for the direction of each effect. NCD, number of chilling days estimated as the number of days between 1 November in the previous year and the LU date with temperatures between 0 and 5 °C; TG, mean growing-season temperature; TP, mean pre-season temperature; SWG, mean growing-season shortwave [visible and near infrared] radiation; LWG, mean growing-season longwave [infrared] radiation; SWP, mean pre-season shortwave [visible and near infrared] radiation; LWP, mean pre-season longwave [infrared] radiation; PG growing-season total precipitation; PP, pre-season total precipitation; DL, day length at LU date; SMG, growing season soil-moisture content; SMP, pre-season soil-moisture content αE, ratio of actual to potential evapotranspiration. drought-prone sites were defined as sites with long-term αE <0.9. Variables were selected based on penalized elastic net regression and corrected for spatial autocorrelation; the coefficient of determination ($R^2$) of the selected models is given at the top of each panel. Error bars represent the 95% confidence interval. The sum of the relative importance equals 100% of R². The scatter plot in each panel represents modeled versus observed variables (LU or GDD$_{req}$) after correcting coefficients for collinearity and spatial autocorrelation. The red line represents the 1:1 line. Source data are provided as a Source Data file.

**a** — LU (ALL)

| | GDDreq | NCD | TG | TP | SWG | LWG | PG | PP | SMG | SMP | αE |
|---|---|---|---|---|---|---|---|---|---|---|---|
| A. hippocastanum | 27.1 | 26.9 | 27.7 | | 12.4 | 1.1 | 3.9 | 0.7 | | 0.2 | |
| A. glutinosa | 34.9 | 31.1 | 24.4 | | 7.6 | 1.6 | | 0.3 | | 0.2 | |
| B. pendula | 26.7 | 27.3 | 29.9 | | 9.1 | 1.6 | 4.3 | 0.2 | | 0.5 | 0.4 |
| F. sylvatica | 29.8 | 28 | 26.2 | | 7.8 | 0.4 | 4.5 | 0.9 | 0.2 | | 2.2 |
| F. excelsior | | 29 | 24.2 | 35.4 | 5.7 | | 3 | | | 0.1 | 2.7 |
| Q. robur | | 25.5 | 29.4 | 30.2 | 7.2 | 1.4 | 4.1 | 2 | 0.1 | | |
| S. aucuparia | 38 | 20 | 27.5 | | 10.3 | 3.4 | | 0.3 | | 0.1 | 0.3 |
| T. cordata | | 2.8 | 42.3 | 34.4 | 11.6 | 3.4 | | 3.2 | | 2.3 | |
| ALL | | 19.8 | 29.6 | 33.6 | 9.5 | 2 | 3.8 | 1.1 | | 0.5 | |

**c** — GDDreq (ALL)

| | NCD | TG | SWG | LWG | SWP | LWP | DL | PG | PP | SMG | SMP | αE |
|---|---|---|---|---|---|---|---|---|---|---|---|---|
| A. hippocastanum | 26.6 | 25.6 | 7 | 1.7 | | 3.9 | 27.1 | 4.5 | 3.7 | | | |
| A. glutinosa | 27.3 | 24.2 | | 3.2 | 31.7 | 5.3 | | 5.6 | 2.6 | | | |
| B. pendula | 25.7 | 31.2 | | 4.4 | 29.2 | 1 | | | 8 | | 0.2 | 0.2 |
| F. sylvatica | 29 | 30.5 | | 4 | 22.3 | 3.5 | | 5.2 | 4.9 | | | 0.5 |
| F. excelsior | 30.5 | 20.2 | | 5.9 | 24.9 | 8 | | 5.6 | 3.5 | 1.4 | | |
| Q. robur | 23.9 | 25.1 | 9.5 | 4.8 | 24.8 | 4.6 | | 2 | | | | 0.2 |
| S. aucuparia | 25.1 | 29.5 | 5 | 5.6 | | 7.6 | 23 | | 3.5 | | 0.2 | 0.4 |
| T. cordata | 7.9 | 39.3 | | 4.9 | 22.3 | 17.1 | | 4 | 4.3 | | 0.4 | |
| ALL | 23 | 26.8 | | 2 | 37.1 | 1.1 | | 5.3 | 4.1 | | 0.2 | 0.2 |

**b** — LU (DROUGHT-PRONE)

| | GDDreq | NCD | TG | TP | SWG | LWG | PG | PP | SMG | SMP | αE |
|---|---|---|---|---|---|---|---|---|---|---|---|
| A. hippocastanum | 30.1 | 27.3 | 29.9 | | | 0.8 | 10.6 | 0.3 | | 0.8 | |
| A. glutinosa | 25.8 | | 26.6 | | 11.1 | 6.3 | 9.3 | 9.4 | 5.9 | | 5.6 |
| B. pendula | 24.3 | 33.8 | 27.8 | | | 0.3 | 1.4 | 9.4 | 2.7 | | 0.2 |
| F. sylvatica | | 27.7 | 22.2 | 29.3 | | 5 | 2.9 | 11.4 | 1.5 | | |
| F. excelsior | | 31.4 | | 40.7 | 13.8 | | 2.4 | | | 2.9 | 8.8 |
| Q. robur | | 23.7 | 26.7 | 30.7 | | 11 | 1 | 6.8 | 0.1 | | |
| S. aucuparia | 46.2 | 27.1 | | | 10.3 | | 10.6 | 3.3 | 2.4 | | |
| T. cordata | | 19.8 | | 38.2 | | 40.2 | | | | 1.9 | |
| ALL | | 22.7 | 26.5 | 36.6 | | 4.4 | 0.2 | 9.2 | 0.3 | | 0.1 |

**d** — GDDreq (DROUGHT-PRONE)

| | NCD | TG | SWG | LWG | SWP | LWP | DL | PG | PP | SMG | SMP | αE |
|---|---|---|---|---|---|---|---|---|---|---|---|---|
| A. hippocastanum | 37.1 | | | 8.7 | 22.6 | 15.6 | | 3.3 | 1.3 | 2.6 | | 8.8 |
| A. glutinosa | 28.3 | | 9.7 | | 29.8 | 9.1 | | 11.6 | 4.2 | | | 7.3 |
| B. pendula | 24.3 | 30 | | 1.9 | 21.8 | 7.2 | | 2.8 | 7.8 | 3.2 | | 1 |
| F. sylvatica | 27.2 | 26.8 | | 9.6 | 16.2 | 8.8 | | 0.1 | 8.6 | 2.3 | | 0.6 |
| F. excelsior | 22.3 | 19.3 | | | 20.2 | 19.1 | | 5.3 | 11.1 | | 2.1 | 0.4 |
| Q. robur | 23.4 | 29.8 | | | 19.1 | 15.2 | | 5.1 | 7.2 | 0.1 | | |
| S. aucuparia | 11.6 | 41.5 | | | 30.5 | 11.6 | | 1 | 2 | 1.7 | | |
| T. cordata | 16.9 | 24.8 | | | 9.9 | 30.2 | | | 11.3 | 2.4 | | 4.5 |
| ALL | 22.3 | 27.2 | | | 32.8 | 9.1 | | 0.2 | 6.5 | | 1.2 | 0.7 |

**Fig. 2** Effect and direction of background climate on LU and GDD$_{req}$ for each species. Relative importance of each variable (in percentage) in explaining leaf unfolding date (LU) and corresponding heat requirement (GDD$_{req}$, estimated as the sum of temperatures > 5 °C between 1 January and the LU date) for each species considering (**a–c**) all sites, (**b–d**) only drought prone sites. The sum of each row is equal to 100% of model R². Red colors indicate a positive correlation, while blue colors indicate a negative correlation. Color intensity reflects variable relative importance. Blanks represent variables that were discarded during the predictors selection. See caption of Fig. 1 for the description of each variable and Table 1 for samples size. The direction and importance of correlations among species is summarized by the size and color of arrows at the bottom of each panel. A double black arrow means that the direction of the response is species dependent. See Supplementary Tables 5–10 in appendix for a complete description of correlation coefficients.

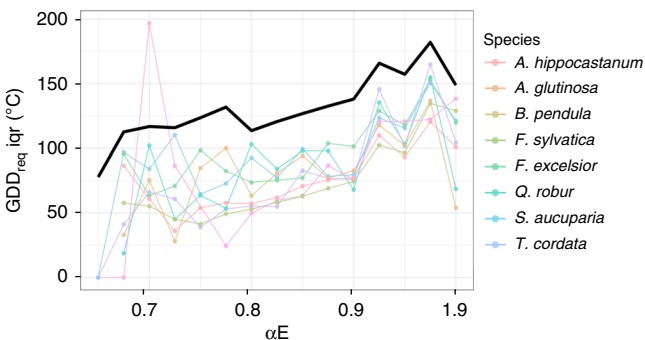

**Fig. 3** Effect of aridity on GDD$_{req}$ spatial variance. Variation of the interquartile (iqr) of the long-term (1970–2016) heat requirement (GDD$_{req}$, estimated as the sum of temperatures > 5 °C between 1 January and the LU date) as a function of the ratio of actual to potential evapotranspiration (αE). Low αE values indicate drought conditions during the growing season. The black line represents the iqr of GDD$_{req}$ when considering all sites and species. See Supplementary Fig. 4 for the distribution of GDD$_{req}$ values along αE. Source data are provided as a Source Data file.

Higher TP implies that GDD$_{req}$ is reached earlier and should thus correlate negatively with LU. This positive correlation thus suggests an adaptation of LU to background temperature.

Even when taking NCD, GDD$_{req}$ and TP into account, which all correlate with pre-season temperature, TG was still selected as the major explanatory variable of LU's spatial variance. Two hypotheses could explain this result: (1) TG is a good integrative proxy of site biogeographical constraints on LU, and potentially summarizes other variables not considered in our study, or (2) trees growing at different locations have optimized the control of their LU in response to biogeographical differences in preseason and growing season conditions[32]. In both cases, this result indicates that adaptation to long-term mean site biogeographical conditions, including growing season conditions but also a suite of biological interactions that could not be included in this study, constitutes an important evolutionary mechanism to optimize LU at that location, and must be considered in addition to commonly used preseason temperatures, which do not suffice to explain the spatial distribution of LU.

**New models are needed for regional studies.** In our study, long-term GDD$_{req}$ was strongly correlated to long-term preseason

temperature (r = 0.95, Supplementary Fig. 6), indicating that GDD models are not independent of sites TP. With constant $GDD_{req}$, higher temperature would imply earlier LU. LU indeed occurs earlier in warm areas and later in cold areas, but much less than expected based on the temporal $GDD_{req}$. Trees have adjusted their $GDD_{req}$ to avoid too late LU in cold areas and risk damage with too early LU at warmer sites. In principle, $GDD_{req}$ should accounts for all warming effects, and therefore be constant unless another driver is at play or adaptation has occurred. So even if LU is partially taken as input to calculate $GDD_{req}$, they would be uncorrelated. The fact that they are correlated highlights that acclimation/adaptation has occurred. The correlation between $GDD_{req}$ and TP and between LU and $GDD_{req}$ are in line with previous remote sensing studies[33,34] and strongly suggest adaptation of spring phenology to long-term site temperature. It also highlights that, even if generally useful to describe the inter-annual variability, $GDD_{req}$ in its current definition is a poor proxy of biogeographical constraints of leaf unfolding.

To emphasize this point, we additionally looked at the commonly used negative relationship between $GDD_{req}$ and NCD. We observed that the relationship between NCD and $GDD_{req}$ for LU is modulated by SWP (Fig. 4). We argue that this relationship is mainly an artifact induced by the fact that we applied uniform $GDD_{req}$ and NCD definition for all sites. Because trees can have different sensitivity to pre-season temperature in different regions (related to light and water availability), it implies that different $GDD_{req}$ and NCD definitions have to be used at the spatial level for different sites if we want to be able to simulate LU at the regional scale. Compared to current regional models using constant $GDD_{req}$ definition independently of the studied region, our results (Figs 2 and 4) suggested that northern sites need to have lower temperature threshold for temperature sum and/or lower $GDD_{req}$ thresholds than southern sites.

Since leaf unfolding determines the restart of the growing season, we expect a requirement/threshold effect of light and water conditions on LU and its temperature sensitivity. Leaf unfolding was also correlated with light and water conditions. The common choice of using GDD models (even when making $GDD_{req}$ dependent on NCD and day length) ignores the effects of the availability of water and light. This partly explains why GDD models typically exhibit large uncertainties when used at regional scales[14,35]. Despite their capacity to well explain the inter-annual variability of LU, the strong correlation between $GDD_{req}$ and long-term mean background climate observed in our study suggests that GDD models, in their current rigid parameterization, are not suitable for studying phenology at the regional scale if they do not include biogeographical constraints, especially for regions where precipitation or light are key controls of LU.

**Radiation intensity matters more than day length for $GDD_{req}$.** Recent experimental spring phenology data have indicated that only 35% of northern hemisphere woody species relied on day length (DL) as a control of LU, and that the dependence on DL occurred predominantly at mid latitudes of the northern hemisphere[36]. Other studies have described a greater impact of day-time than nighttime temperatures on LU date[37,38] as well as differences in $GDD_{req}$ and NCD requirement depending on DL[11]. Day length and LU date are highly collinear (Supplementary Fig. 6), making it difficult to separate the impacts of day length (or preseason radiation) on LU. However, several previous studies did suggest a modulation of the temperature sensitivity of LU by light[11,36–38]. In our study, growing and preseason radiation intensity received by plants were clearly identified as an important explanatory variable of $GDD_{req}$'s spatial variance (Figs 1 and 2) and were more important than day length in explaining the spatial differences in $GDD_{req}$.

While the effect of light intensity was clear, contributions to the spatial heterogeneity of LU and $GDD_{req}$ were very different between SWP and LWP. The ratio of visible to infrared light

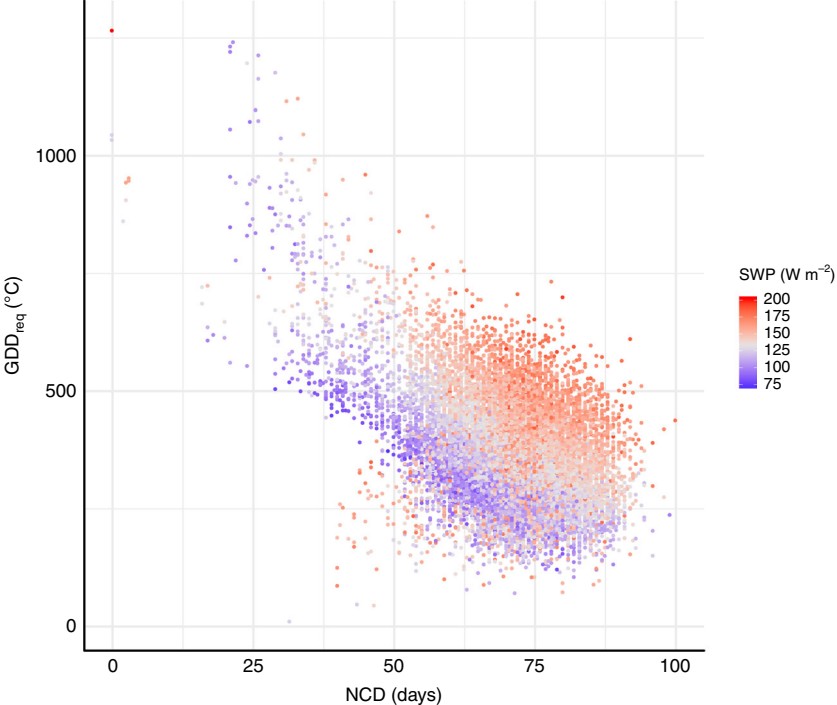

**Fig. 4** Adaptation of chilling-forcing requirement to preseason radiation. Scatterplot of average site long-term (1970–2016) $GDD_{req}$ (°C) versus NCD (days) for all species and all sites. The color gradient represents the average site long-term incoming shortwave preseason radiation (SWP, W m$^{-2}$). Gray color represents the median value of SWP (130 W m$^{-2}$) of all sites, blue and red colors correspond to values lower and higher than the median SWP, respectively. Source data are provided as a Source Data file.

determines several plant processes, such as the state of phytochrome photoequilibrium, which controls growth rate, foliar and chloroplast development and even apical dominance[39]. Some evidence also suggests that light modulates internal hormone-regulated growth[40] and protein production in plants[41] by affecting signaling pathways of ethylene and abscisic acid, two phytohormones involved in bud set and leaf development[42]. A recent experiment found a phytochrome-mediated photoperiodic control for *Fagus sylvatica*[43], while a recent review highlighted the effects of the light spectrum on spring and autumn phenology[44]. We therefore hypothesize that trees adjusted their life cycle to the average light spectrum of the site that can be directly sensed by buds and plays a direct role in enhancing or inhibiting LU, but this remains to be experimentally verified also for other species.

Light plays a key role in plant activity during the growing season by regulating photosynthesis and growth. The growing season is also the period during which buds are created. In the long-term, we expect that trees respond to growing season meteorological conditions during which the formation of buds, but also carbon reserves, can be affected, which in turn affects the sensitivity of buds to temperature in the subsequent winter and spring as already shown with *Populus tremula*[45]. However, differences in stand canopy openness, leaf area or even plant activity can lead to large uncertainties in the potential effect of growing season radiation on tree eco-physiology due to differences in local light regimes. As for TG, both SWG and LWG may be good proxies of background biogeographical constraints without having a direct effect on phenology.

We argue that intensity of growing and preseason incoming radiation should be included in phenological studies, not only day length as a proxy of photoperiod. More research, especially experimental studies, is needed to clearly distinguish among the effects of the light spectrum, light intensity and day length on LU and its required $GDD_{req}$.

**The role of aridity: The water-saving hypothesis**. No theory has yet been accepted that accounts for the effect of water on spring foliar phenology in temperate forest ecosystems. Here, we propose a hypothesis on the effect of drought stress on LU dates and $GDD_{req}$. LU and $GDD_{req}$ were more strongly correlated to water availability at drought-prone sites (Fig. 2) than at wetter sites, potentially reflecting a long-term adaptation of trees to frequent drought stress. Temperature sensitivity tends to be lower in water-limited conditions, as indicated by the higher $GDD_{req}$ with increasing drought stress (Fig 2 and Supplementary Fig. 4). Organogenesis and primary growth in buds have been correlated with hydraulic architecture[46], and previous studies have highlighted a clear effect of growing-season water stress on bud production and foliar development[47,48]. Here, we highlight an additional effect on the sensitivity of buds to temperature in spring.

We speculate that bud acclimation to previous drought may represent a water-saving strategy. By decreasing the bud sensitivity to TP, trees delay LU and the associated start of evapotranspiration[49]. The identification of preseason precipitation as an important control of the spatial variance of LU at drought-prone site is in line with this hypothesis and might also reflect a safety mechanism by which plant delay leaf unfolding until water is available for the restart of plant activity. As a long-term strategy, a delay in evapotranspiration will lead to a slower depletion of water resources at the beginning of the growing season and reduces the risk of water stress during late spring and summer when radiation is more favorable for photosynthesis.

**Species show different responses to long-term constraints**. All species exhibited the same response to temperature, however a few species responded differently to water availability at drought-prone sites. *A. glutinosa*, *T.cordata* and *F.excelsior* showed opposite correlation with water availability compared to the other species (Fig. 2). *T. cordata* and *F. excelsior* were proposed as species with a relatively high drought tolerance compared to other European species[50,51], while *A. glutinosa* naturally occurs in wet sites. This different behavior observed for *A. glutinosa*, *T.cordata* and *F. excelsior* could reflect either their drought tolerance, or the fact that differences in edaphic heterogeneity were not captured by aggregating data at the pixel level, resulting in a mismatch between actual water availability and the estimates of αE used here.

Differences were also observed for some species regarding their heat requirement. Biogeographical conditions of temperature, light and water only captured 33, 44 and 49% of $GDD_{req}$ spatial variance for *B.pendula*, *F. sylvatica* and *A. glutinosa*, respectively. Previous studies showed a reduced sensitivity of leaf unfolding to climate warming in the last decades, mainly attributed to plant plasticity[52]. However, how trees acclimate or adapt to future climate change remains unclear and might be species-dependent, due to differences in temperature, light and water sensitivities.

The remaining unexplained variance of LU and $GDD_{req}$ can be attributed to uncertainties in observations and climate data, with a potential effect of data aggregation at the pixel level, but also of unaccounted biotic and edaphic factors (e.g., stand age, biogeography of pathogens or mycorrhizal associations, soil structure and fertility, etc.).

In the end, the co-limitation of spring phenology by light and aridity may account for why LU does not keep pace with climate change[19,52], which may have vast and far implications on the carbon cycle[53], with a possible alteration of the competitive balance among species[54].

**Tree seasonality affects spring phenology**. Temperature, availability of light and water and their interactions thus affect the spatial heterogeneity in LU date and its associated $GDD_{req}$. On one hand, our results showed that LU date adjusted to spring meteorological conditions, for which trees are co-limited by TP, SWP, LWP and PP. On the other hand, we observed different responses of spring phenology to preseason and growing-season meteorological conditions, especially visible for drought-prone sites, highlighting that tree adjust their phenology to cope with seasonality on the long-term.

Since buds are formed during the growing season, we argue that any effect of growing season pressure might in turn have a feedback on tree seasonality, with a potential long-term adjustment of ecophysiological responses to these constraints. In the long term, it may also alter the restart of plant activity in spring and thereby optimize the long-term growth, reproduction, or survival of the trees, which influence the restart date of physiological processes.

The impact of elevated air temperature during the growing season has been proposed to affect spring leaf unfolding by modifying growth cessation and dormancy induction in temperate and boreal trees[55]. Drought stress also affects the onset of senescence and possibly dormancy[56,57]. Precipitation has indeed been shown to play a large role in vegetation dynamics during the senescence period of deciduous forests in the Northern Hemisphere[58]. A delayed dormancy induction can translate into a delayed leaf unfolding during the next spring via the later start date of chilling and GDD accumulation[59]. Our results are fully in line with these observations and provide additional evidence that the timing of phenological events is impacted by previous phases in the annual cycle of trees.

We argue that tree seasonality and long-term biogeographical constraints are too often overlooked and should be taken into

account in phenological studies. Further research on the effect of tree seasonality on inter-annual variability of phenology is needed to clearly identify the role played by biogeographical constraints.

In conclusion, assessing the long-term spatial variance of LU and $GDD_{req}$ is a step in developing a unified framework that will allow an understanding of the multiple control of climate on plant phenology. Future research on the importance of plant phenology on ecosystem functioning should focus on space-time interactions with environmental conditions specifically to address: 1) the effects of light and aridity on bud sensitivity to temperature, and 2) the potential coordination between plant processes and phenology that could account for a co-limitation by temperature and the availability of light and water. In line with these recommendations, the use of current, constant, $GDD_{req}$ and NCD metrics for the study of spatio-temporal patterns in plant phenology should be used with caution.

## Methods

**Datasets**. Data for in situ leaf unfolding (LU) were obtained from the Pan European Phenology (PEP) network (www.pep725.com) and the GDR2968 database (http://www.gdr2968.cnrs.fr/) for France. Phenological observations followed the Biologische Bundesanstalt, Bundessortenamt und Chemische Industrie (BBCH) code, with LU corresponding to BBCH = 11.

Daily temperature, precipitation and incoming radiation (both shortwave [visible and near infrared] and longwave [infrared]) were retrieved from the CRU-NCEP climatic data set[30] at a spatial resolution of 0.5°. Day length was calculated for each site using the 'geosphere' R package[60]. Monthly SM (10–40 cm) was retrieved from the land data assimilation systems (GLDAS) data set (https://ldas.gsfc.nasa.gov/gldas/) aggregated at a spatial resolution of 0.5° to be consistent with the CRU-NCEP data.

We approximated drought conditions using the ratio of actual to potential evapotranspiration as a proxy for drought periods (αE). This ratio accurately represents droughts[61] and was calculated using the SPLASH model[31]. αE was calculated daily, but only the growing-season αE was used to calculate long-term drought stress.

**Analyses**. We corrected site temperature for altitudinal differences between the site and the mean elevation of each grid cell of the CRU-NCEP data set[14] using a gradient of 6.4 °C km$^{-1}$ and then estimated the heat requirement and associated chilling for each LU observation. Heat requirement ($GDD_{req}$) corresponds to the sum of mean daily temperatures above a threshold of 5 °C calculated from 1 January to the date of LU and was calculated for each observation as:

$$LU = d(GDD_d = GDD_{req}) \qquad (1)$$

$$GDD_d = \sum_{t_0}^{d} \max((\bar{T}_d - T_{th}), 0) \qquad (2)$$

$GDD_{req}$ defines the heat requirement of buds at the observed LU date, estimated as the accumulated daily air temperature ($GDD_d$), where $\bar{T}_d$ is the mean daily air temperature, $T_{th}$ the temperature threshold for GDD accumulation (5 °C) and $t_o$ the starting date (1 January).The number of chilling days (NCD) was calculated as the number of days from 1 November to the LU date with mean daily temperatures between 0 and 5 °C.

We were interested in the spatial variance of LU and the heat requirement, so we used the median LU date and the corresponding median GDD for 1970–2016 for each site, assuming that the medians represented the long-term "optimal" LU date and GDD for the background climatic and soil conditions. We only retained sites with more than 5 years of observations and removed years with a LU date outside two interquartiles around the median distribution (i.e., outside days 80–152), which potentially represent a response to extreme events. The same analysis was performed using sites with more than 10 years of observations and led to similar results. We analyzed the correlations between biogeographical variables, LU dates and GDD for eight species of dominant European deciduous trees with many records in the PEP725 database: *Aesculus hippocastanum*, *Alnus glutinosa*, *Betula pendula*, *Fagus sylvatica*, *Fraxinus excelsior*, *Quercus robur*, *Sorbus aucuparia* and *Tilia cordata*.

Long-term climatic variables (i.e., over 1970–2016) were averaged for two periods of the year: the growing season, defined as the period between days 180 and 250, and the preseason, defined as the three months before LU. Since we are interested in biogeographical constraints, not the temporality of processes, and because no information about the length of the growing season was available in the PEP data set, selecting a constant summer period ensures that we have a representative period for all trees and all years. It allowed a consistent comparison between sites without introducing bias induced by different growing season lengths. This period also corresponds to the peak of plant activity in temperate

ecosystems, and over which we are more likely able to gather information about water and temperature pressures of each site.

As for LU, extreme climatic years were excluded as we seek to estimate the average response of the vegetation. We then analyzed the spatial relationships between long-term climatic variables (averaged over the different periods of the year), LU and GDD. We proceeded in four steps:

1. We first assessed potential collinearity issues between variables by examining pairwise correlation coefficients[62] (Supplementary Fig. 6);

2. We then selected relevant predictors of observed LU and $GDD_{req}$ using penalized elastic net regressions (glmnet function from the glmnet[63] package) in combination with collinearity information from step 1). Interaction terms were not included in the analysis. In step 2–4) predictors were all standardized in order to represent the relative contribution of each variable in explaining LU and GDD variability (Supplementary Tables 8 and 9).

3. After selecting relevant variables in step 2, we assessed the remaining collinearity between variables by estimating their respective variance inflation factor (VIF) as:

$$VIF_j = \frac{1}{1 - R_j^2} \qquad (3)$$

where the VIF for variable j is the reciprocal of the inverse of R² from the regression. VIF values increase with collinearity, and arbitrary threshold of 5–10 are commonly used to define high VIF values. Here, when two predictors exhibited potential collinearity (i.e., high VIF values), we removed the one with a VIF value higher than 4 and the lowest correlation coefficient with LU or GDD using the stepVIF function from the VIF[64] package.

4. We finally assessed the spatial structure of residuals of the reduced model from step 3) using semi-variograms (Supplementary Figs 7–9). We performed generalized least square regressions taking into account the spatial structure of residuals to correct coefficients using the gls function from the nlme[65] package. Different spatial structures were tested (linear, exponential, spherical, Gaussian and rational quadratic) and the best model was selected using AIC criterion (Supplementary Table 10).

All the above steps were applied to each species separately. Because results were consistent between species (Fig. 2) regarding aggregation uncertainties and because we aimed at exploring biogeographical constraints at the regional scale for all deciduous tree species, we applied the same approach to the full dataset with all species pooled together. Analyses were performed with the R v.3.5 software[66].

**Reporting summary**. Further information on research design is available in the Nature Research Reporting Summary linked to this article.

## Data availability
All phenology data are available at http://www.pep725.eu/ and http://www.gdr2968.cnrs.fr. CRU-NCEP data can be downloaded at https://rda.ucar.edu/datasets/ds314.3/. The SPLASH model used to estimate evapotranspiration can be downloaded at https://bitbucket.org/labprentice/splash/. Soil moisture data can be downloaded at https://ldas.gsfc.nasa.gov/gldas/. The source data underlying Figs 1, 3, 4 and Table 1 and Supplementary Figs 2 and 4 are provided as a Source Data file.

## Code availability
No custom code nor mathematical algorithms were developed for this study. Only existing packages and software were used for the analysis, which can be found in the Methods.

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

## Acknowledgements

We acknowledge the financial support from the European Research Council Synergy grant ERC-SyG-2013–610028 IMBALANCE-P. We acknowledge all members of the PEP725 project for providing the phenological data. B.D.S. was funded by the Swiss National Science Foundation grant no.PCEFP2_181115. M.P. acknowledges financial support from the Fonds WetenschappelijkOnderzoek (FWO) grant G018319N.

## Author contributions

M.P., I.J. and J.P. designed the study. M.P. performed the analyses and wrote the first version of the manuscript. B.D.S. provided evaporation data. R.M.H. contributed to statistical analysis. M.P., I.J., B.D.S., A.D.F., Y.H.F., R.M.H., P.C. and J.P. contributed to the interpretation of the results and to revisions of the manuscript.

## Competing interests

The authors declare no competing interests.

## Additional information

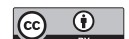

