## [Peer Review File · Nature Communications]

Reviewers' comments:

Reviewer #1 (Remarks to the Author):

This study investigated which environmental drivers influence the spatial variation of leaf unfolding in 8 temperate deciduous tree species in Europe. The median leaf unfolding date from 1970-2016 was assumed to represent long-term optimal leaf unfolding date for each site and was compared to potential (1) temperature, (2) radiation, and (3) precipitation/aridity drivers (using means from 1970-2016). Multiple drivers are included for each of these three categories, but there is redundancy within the set of 13 variables that could be removed (or otherwise, it should be defended).

Temperature was the most important factor controlling spatial variance in leaf unfolding for all species, but radiation and precipitation/aridity also had significant effects. The most novel finding is that leaf unfolding differences existed between wet and dry temperate forest sites – radiation and precipitation were more important at drier sites. Radiation was more important than daylength in describing spatial variation in GDD within dry sites. There is no direct mention of differences between warm versus cold sites, although differences seem to exist (Fig. 2d, 2f). Also, the authors spend little time discussing species differences, despite some interesting patterns emerging from their analyses (Fig. 2b, 2d, 3e).

Introduction:

Line 51-52: "Studies have mostly focused on the temporal (interannual) changes in spring phenology, but..." Adding more specific detail here could help the reader to better understand the different focus of your study, i.e. 'Many studies have documented that recent warming trends have advanced leaf unfolding dates for many temperate tree species, but ...'

Line 59: "mitigates"

Lines 62-64: The term "acclimation" has a very specific meaning to plant ecophysiologicalists, which may be different than your meaning here. I consider an acclimation response as occurring over days to months, allowing organisms to maintain homeostasis in response to environmental change, whereas adaptation requires decades to centuries (see Figure 1 in Smith and Dukes, 2013, Plant respiration and photosynthesis in global-scale models: incorporating acclimation to temperature and CO₂, Global Change Biology). Basically, are you meaning temporal variance at an interannual time scale? If so, within how many years? Clarification may require additional discussion of how quickly species can adapt their phenology to persistent warming trends.

Lines 64-70: This example does a good job of highlighting the complexity of environmental differences between high and low elevations, i.e. potential factors affecting spatial variation in

phenology, but it doesn't help to clarify your theory for different mechanisms controlling temporal versus spatial variation. Could you expand this to better explain why temporal variation in spring phenology would be a short-term acclimation response? Rather, I have always interpreted temporal variation in spring phenology as an aspect of long-term adaptation to a temperate climate. For example, isn't delayed budburst in a cold year (relative to a warm year) an adaptation to avoid frost damage?

Line 80-81: Clarify, is NCD where the maximum or mean daily temperature is 0-5 C? (Actually, your methods indicate that it's mean, but it would be good to indicate here too.)

Line 89-90: It would be helpful to include several other details here: Are the tree species all deciduous? What is your definition of "long-term", i.e. what dates? What geographic spatial extent did you analyse? How many sites were included in your analyses?

Line 95: day of year, DOY

Methods:

Line 314-318: A warming trend did exist over your study period (1970-2016), but given that you are using the median LU over this nearly 50-year period, this seems logical reasoning. I agree that it is appropriate to exclude responses from extreme years, but why didn't you also exclude extreme years when calculating mean climate variables?

Lines 324-338: It seems redundant to have preseason (TP) and growing season temperature (TG) in the analysis (also GDD), for several reasons. There is likely a strong correlation between these variables across study sites. Further, growing season temperature occurs after leaf unfolding, by definition, and thus is not a likely candidate for a phenological cue. Also, inclusion of many variables makes interpretation more difficult. The same argument could be used for other variables, including radiation and precipitation. While the nature of your study is not necessarily focused on environmental cues for budburst phenology, but rather evolutionary pressures on leaf phenology, this important distinction was not made clear in the introduction.

Results:

Your overall structure here does not do the best possible job of highlighting your main results, plus you return to explaining temperature effects under the "Precipitation" heading. Rather than using "Temperature", "Radiation", and "Precipitation", you could split between "Wet temperate forest" and "Temperate forest with more frequent drought". Were there any differences between warm versus cold sites?

Figure captions: How many sites are included in analysis? It is okay to indicate a range of numbers to account for differences among species, but this is important information to describe.

Figures 2-4: Please add in a brief description in the caption of what negative and positive numbers indicate. The interpretation is difficult for me and may be confusing for other readers too.

Lines 108-110: As mentioned previously, it is strange to make a distinction between GDD, TP, and TG. These are all temperature-based measures that are likely to be correlated across sites. Rather, I expected to read here about the contribution of radiation and precipitation.

Lines 115-118: This is interesting. Differences among species could be indicative of different adaptations that have resulted from evolutionary pressures by these environmental variables. I recommend adding more information to shift your results to this focus. For example, how much variation exists among species in leaf unfolding date and GDD, within a site? How does the range in LU/GDD across sites differ among species? Are all species found at all sites, or is the distribution of some species restricted to certain sites (i.e. wet sites)? Which species are considered more drought tolerant (e.g. *Fagus* and *Quercus* are generally considered to be more drought tolerant than *Tilia* and *Sorbus*) and does this help to explain results?

Lines 158-159: In a global context, arid ecosystems are generally considered to have very little annual rainfall (<300 mm), e.g. deserts, drylands, etc. Therefore, I suggest changing your use of "arid" in table/figures to "drought-prone" or "drier" or "dry" to avoid confusion and fit your definition better. Or if you really wanted to examine only the driest sites, maybe you should consider using a lower AET:PET threshold than 0.9? For example, the largest differences among species in Figure 4d occur at AET:PET<0.8. Also, please state how many sites were characterised as "arid", at least in table/figure captions.

Discussion:

Lines 192-207, 236-244: Much of this section reads as if you are describing evidence for environmental cues of budburst phenology. Is this really the main focus of your analysis?

Lines 210-213: But didn't your analyses pair responses to climate in the same year, not the previous year?

Line 223: Cite relevant figures here, and throughout discussion.

Lines 228-229: This hypothesis was not mentioned in the introduction. It would be helpful to set up these findings by describing potential hypotheses at the beginning. See also lines 205-206.

Lines 229-230: Clarify if you are referring here to intraspecific or interspecific variation. Are there any relevant species differences that could be described more specifically and referred to here?

Lines 247-248: But drought can also lead to early leaf senescence in some species, particularly those with low drought tolerance (see Marchin et al. 2010, Drought-deciduous behavior reduces nutrient losses from temperate deciduous trees under severe drought. *Oecologia*, 163: 845-854).

Conclusion:

Lines 271-272: Isn't conclusion #2 only applicable to dry sites? Table 1 indicates that both radiation and daylength were only included in the GDD model for dry sites. Please clarify to prevent overstating your findings.

I hope that you find these comments helpful,

Renée M. Marchin

Western Sydney University

Reviewer #2 (Remarks to the Author):

General comments:

Marc Peaucelle and colleagues combine a large-scale European dataset of spring phenology from eight tree species with several climate variables to assess biogeographical effects of temperature, light and aridity on the timing of leaf unfolding (LU) and on growing degree days (GDD). For both LU and GDD, the long-term median from 1970 to 2016 was used at each site, for the climate variables, long-term means from 1970 to 2016 were used. While the spatial extent and the large amount of the data is truly impressive, I think, the study lacks scientific rigour to be published at the current stage. I want to briefly summarize the major concerns that I encountered:

While the study has been framed into a spatial context, there is apart from Supplementary Fig. 2 no spatially explicit analysis of the data. Rather than treating the spatial observations as independent, which was assumed for the linear models and the partial correlations, spatial autocorrelation needs to be accounted for. Otherwise, an important assumption of the linear model is violated. Treating spatially autocorrelated observations as independent results in biased standard errors, P values and R^2 values, thus significant effects are found more often than there actually are. There

are many ways how to deal with spatial data (see e.g. Dormann et al. 2007). The “nlme” package in R provides several options to model spatial correlation structures (see Pinheiro and Bates 2000).

The multiple linear models to predict LU date and GDD (Table 1) include not only long-term mean climatic variables from the pre-season (i.e. three-month period before LU) but also from the growing season (i.e. period from days 180 to 250). I was puzzled to see that climatic variables such as the mean growing-season temperature (TG), the mean growing-season short- and longwave radiation (SWG, LWG), the growing-season total precipitation (PG) and the growing-season soil-moisture content (SMG) were used to predict a phenological event (LU) that actually occurs at the beginning of the growing season (i.e. either prior to the observation period of these growing-season variables, or only in spring of the following year). While the authors tried to interpret these effects in the Discussion (L235-253), i.e. how the formation of buds during summer and their sensitivity to weather variability affects spring phenology, the current knowledge of phenological processes does not support these results. On the one hand, logic reasoning rejects the possibility of retrospective effects of growing-season climate variables on spring phenology in the current year. On the other hand, if the climatic conditions during the current growing season (i.e. mainly July and August) had such a strong impact on spring phenology of the following year (i.e. more than 6 months following the predefined growing season), this would be known for a long time. Common knowledge, however, suggests that the onset of leaf unfolding is mainly driven by forcing temperatures in spring that interact with chilling temperatures in winter.

The main issue with these analyses is that correlation does not imply causality. Many of the effects that are shown in Table 1 or Fig. 1 occur because of the high correlation among variables. While at a warm site not only high spring temperatures are observed resulting in relatively early budburst, but also high summer temperatures, whereas at a cool site spring temperatures are rather low that result in delayed budburst, followed by cool summer temperatures. To detect potentially lagged effects of growing-season climate variables on leaf-out time or GDD in the following spring, site-specific models need to be used. Thus, instead of using long-term mean climate variables, the effects of climate variables from the current growing period on LU date or GDD of the following year need to be estimated.

I was further surprised to see (1) that temperature and precipitation during the growing season are suggested to have almost the same relative importance as GDD, and (2) that the number of chilling days (NCD) has actually the highest relative importance for explaining LU date (see Fig. 1a). For most species NCD seems to be more important than GDD for predicting LU, and for all species the mean growing-season temperature (TG) is more important than GDD (Supplementary Fig. 4). If these findings were true, they would question most phenological studies that have been previously published. However, a majority of mechanistic phenology models does not even show any significant contribution of chilling on leaf-out time (not to mention of growing-season variables of the previous year), i.e. the main driver seems to be forcing in spring (e.g. GDD; see e.g. Basler 2016).

Specific comments:

L64-70: The sentence should be split up into two sentences.

L67, L121 etc.: Replace "altitude" with "elevation" (see McVicar and Körner 2013).

L88-93: Forest trees are affected by local light conditions, which can strongly deviate from large-scale conditions. Thus, the local light regime is probably more important than the overall light regime (compare e.g. a free-standing tree at high latitude with a tree growing under a dense canopy at low latitude.).

L89: I suggest to define "long-term" also in the main text, otherwise this only gets clear when reading the methods (L314).

L95, L325-328: Days 180 to 250 probably reflect the growing period at just a few sites (high latitudes or high elevations). However, at many sites the growing period starts several months before and ends 1-2 months afterwards. E.g. the mean temperature calculated over the actual growing period can substantially differ from the mean temperature over days 180 to 250. Thus, mean values from day 180 to 250 do not generally represent conditions during the growing season.

L99: Unclear why the subtitle is "temperature". Other variables not related to temperature are presented as well, e.g. radiation or water availability. These results should be moved to "Radiation and day length" and "Precipitation and aridity", respectively.

L100: Stepwise regression should not be used because of various shortcomings (e.g. p. 56 in Harrell 2001), instead information-theoretic model selection should be used (e.g. Stauffer 2008).

L104: The R^2 values are biased because of spatial autocorrelation (see e.g. Dormann 2007 or Lennon 2000).

L109: What is the causality between growing season temperature (TG) and LU? If growing season variables show an effect, this is because of correlations with pre-season variables (e.g. at a warm site, temperature is high during the pre-season as well as during the growing season).

L113: Standard errors need to be provided for all slope estimates.

L125-127, L146-147, L156-157: Here, too, the question of causality arises.

L129, L145-151 etc.: I recommend to provide exact P values (not just significance levels).

L131-132: However, the effect of SWP and LWP on LU was not tested, because the variables were not included in the models (see Table 1, L554-555).

L144-145: However, the effect seems to be significant ($p < 0.05$). Just because the estimated coefficient is small does not mean that LU date is not sensitive to PG. The coefficient depends on the scale on which PG is measured.

L183: However, in temperate forests, forcing (e.g. GDD) is generally more important than chilling, because chilling conditions are fulfilled related to the rather cool winter conditions.

L184-185: How can this effect be explained? TG represents the mean growing season temperature (e.g. a variable that is measured after observing LU).

L210-213: If this was the case, the site-specific effects of radiation need to be quantified. Thus, the effect of radiation during the current growing season on LU date and GDD in the following year needs to be assessed. The current models do not allow to draw these conclusions because long-term variables are used.

L242-244: I guess this is a purely correlative effect: at a warm site, temperature is not only higher during the growing season but also during the pre-season.

L314: However, the date of leaf unfolding may differ by 1-2 months at a specific site, i.e. there is large variability that is not accounted for with the median.

L335: Write "collinearity".

Figure 1: (1) Why is NCD more important than GDD? If most observations were from warm sites where chilling requirements are often not fulfilled, I might be inclined to believe this. However, many sites are from areas with a clearly seasonal climate (i.e. alternation between rather warm summer and cool winter). (2) E.g. the relative importance of TG on GDD is of purely correlative nature (i.e. both mean growing-season temperature and GDD at warm sites are higher than at cool sites).

Figure 2: Write “residuals” (y axis label).

Figure 5: It seems that there is an offset, i.e. the black line should be shifted down to actually represent the trends for all sites and species.

Table 1: (1) It should be explained why some variables were not included in the model (L554-555 and Supplementary Figure 3). (2) Standard errors / confidence intervals and exact P values need to be provided. Just based on the coefficients, it does not get clear, which variables are important and what the uncertainty of the estimates is. (3) LU and GDD are expressed by TG, SWG, LWG, PG, and SMG, i.e. values that are measured after LU or more than 6 months before LU in the following spring (see my previous comments about causality). (4) The spatial autocorrelation of the data is likely to result in biased R2 values.

Supplementary Figure 4: The variables SWP, LWP and DL are mentioned in the legend, but are not shown in the figure. A brief explanation for GDD is missing in the legend. R2 values are not shown in the figure.

Supplementary Figure 5: The variables TP and DL are not shown in the figure. R2 values are not shown in the figure.

Supplementary Figure 7: R2 values are not shown in the figure.

References:

Basler D. 2016. Evaluating phenological models for the prediction of leaf-out dates in six temperate tree species across central Europe. *Agricultural and Forest Meteorology* 217:10-21.

Dormann C.F. 2007. Effects of incorporating spatial autocorrelation into the analysis of species distribution data. *Global Ecology and Biogeography* 16:129-138.

Dormann C.F., McPherson J.M., Araújo M.B., Bivand R., Bolliger J., Carl G., Davies R.G., Hirzel A., Jetz W., Kissling W.D., Kühn I., Ohlemüller R., Peres-Neto P.R., Reineking B., Schröder B., Schurr F.M. and Wilson R. 2007. Methods to account for spatial autocorrelation in the analysis of species distributional data: a review. *Ecography* 30:609-628.

Harrell F.E. 2001. Regression modeling strategies. Springer Verlag, New York, Berlin, Heidelberg.

Lennon J.J. 2000. Red-shifts and red herrings in geographical ecology. *Ecography* 23:101-113.

McVicar T.R. and Körner C. 2013. On the use of elevation, altitude, and height in the ecological and climatological literature. *Oecologia* 171:335-337.

Pinheiro J.C. and Bates D.M. 2000. Mixed-effects models in S and S-PLUS. Springer, New York.

Stauffer H.B. 2008. Contemporary Bayesian and frequentist statistical research methods for natural resource scientists. Wiley and Sons, Hoboken, New Jersey.

Reviewer #3 (Remarks to the Author):

The authors present a study using the PEP phenology database to analyze spatial climate controls on phenological sensitivity to a variety of important variables (e.g., GDDs, light, chilling etc). Overall this is an interesting and detailed study, but I have some significant methodological concerns that I feel need to be addressed.

The first, and most important in my mind, is the multiple stepwise linear regression results (Table 1 and Figure 1). Specifically, there is significant collinearity across all of the predictor variables that to me suggest these models are way overfit. For example, sites with warmer growing and pre-season temperatures are likely to also be sites with greater growing degree days and lower chilling. But it's not even just the temperature variables alone. Because both temperature and solar radiation both typically decline with latitude, these variables are also likely to be significantly correlated. Dry sites as well are also typically warmer in this region of the world, and so again moisture and temperature will not be independent. Nowhere, however, can I find any evidence that the authors have addressed these issues and, indeed, I'm not even sure it is possible given the largely inseparable interactions between these variables. Maybe this is dealt with in the partial correlation analysis, but it's unclear how this translates to the multiple regression, which by all accounts does not address collinearity issues.

Second-I don't understand why the growing season predictors are even included, given that these metrics are dominated by things happening AFTER leaf unfolding. Why, then, should we even consider them in the first place? I guess the argument is that the impact from this season would

manifest over evolutionary timescales, but it still feels like a bit of a weak argument without other evidence to back it up.

Reviewers' comments:

Reviewer #1 (Remarks to the Author):

[Comment #R1.1] This study investigated which environmental drivers influence the spatial variation of leaf unfolding in 8 temperate deciduous tree species in Europe. The median leaf unfolding date from 1970-2016 was assumed to represent long-term optimal leaf unfolding date for each site and was compared to potential (1) temperature, (2) radiation, and (3) precipitation/aridity drivers (using means from 1970-2016). Multiple drivers are included for each of these three categories, but there is redundancy within the set of 13 variables that could be removed (or otherwise, it should be defended).

[Response #R1.1] Following the reviewer's suggestion and comments, we updated the main text and the methodology to justify the use of both pre-season and growing season variables while checking for redundancy, collinearity and spatial auto-correlation issues:

- First, we used multiple climatic drivers representing different phases of the life cycle of trees, because recent evidences suggest that tree seasonality plays a key role on eco-physiological processes (Le dantec et al., 2000, Breda et al. 2006; Hänninen & Tanino, 2011, Fu et al., 2014; Keenan and Richardson 2015; Strømme et al., 2015; Liu *et al.*, 2018). One of the main hypotheses, that we seek to explore with this study, is that tree phenology is constrained not only by spring meteorology as commonly suggested, but also by background biogeographical conditions. We argue that confronting pre-season and growing season conditions is more representative of biogeographical constraints than using spring conditions only. We revised the introduction section (Line 65-75 & L.97-109) to highlight this point. (see also Response #R1.12).

- We also revised the “Methods” section (Lines 422-454) as follow:

1) We first assessed potential collinearity issues between variables using a correlation coefficient as suggested in Dorman *et al.* (2013);

2) We then selected relevant predictors using penalized elastic net regressions in combination to collinearity information from step 1). In steps 2-4) predictors were standardized in order to represent the relative contribution of each variable in explaining LU and GDD variability.

3) After selecting relevant variables in step 2, we assessed the remaining collinearity between variables by estimating their respective generalized variance inflation factor (VIF) as :

$$VIF_j = \frac{1}{1 - R_j^2}$$

where the VIF for variable j is the reciprocal of the inverse of R^2 from the regression corrected for the degree of freedom. VIF values increases with collinearity and arbitrary threshold of 5-10 are commonly used to define high values. Here, we removed variables with VIF higher than 4 using a backward stepwise procedure.

4) We finally assessed the spatial structure of residuals of the reduced model from step 3) with semi-variograms. In case of spatial auto-correlation, we performed generalized least square regressions taking into account for the spatial structure of residuals to correct coefficients. Different spatial correlation structures were tested (linear, exponential, spherical, gaussian and rational quadratic), and the best model was selected using AIC criterion.

All the above steps were applied to each species separately and to the full dataset with all species pooled together.

Hänninen, H. & Tanino, K. Tree seasonality in a warming climate. *Trends Plant Sci.* **16**, 412–416 (2011).

Le Dantec, V., Dufrêne, E. & Saugier, B. Interannual and spatial variation in maximum leaf area index of temperate deciduous stands. *For. Ecol. Manag.* **134**, 71–81 (2000).

Strømme, C. *et al.* UV-B and temperature enhancement affect spring and autumn phenology in *Populus tremula*. *Plant Cell Environ.* **38**, 867–877 (2015).

Bréda, N., Huc, R., Granier, A. & Dreyer, E. Temperate forest trees and stands under severe drought: a review of ecophysiological responses, adaptation processes and long-term consequences. *Ann. For. Sci.* **63**, 625–644 (2006).

Fu, Y. S. *et al.* Variation in leaf flushing date influences autumnal senescence and next year's flushing date in two temperate tree species. *Proc. Natl. Acad. Sci.* 201321727 (2014).

Keenan, T. F. & Richardson, A. D. The timing of autumn senescence is affected by the timing of spring phenology: implications for predictive models. *Glob. Change Biol.* **21**, 2634–2641 (2015).

Liu, G., Chen, X., Zhang, Q., Lang, W., & Delpierre, N. (2018). Antagonistic effects of growing season and autumn temperatures on the timing of leaf coloration in winter deciduous trees. *Global change biology*, 24(8), 3537-3545.

Dormann, C. F., Elith, J., Bacher, S., Buchmann, C., Carl, G., Carré, G., ... & Münkemüller, T. (2013). Collinearity: a review of methods to deal with it and a simulation study evaluating their performance. *Ecography*, 36(1), 27-46.

[Comment #R1.2] Temperature was the most important factor controlling spatial variance in leaf unfolding for all species, but radiation and precipitation/aridity also had significant effects. The most novel finding is that leaf unfolding differences existed between wet and dry temperate forest sites – radiation and precipitation were more important at drier sites. Radiation was more important than daylength in describing spatial variation in GDD within dry sites. There is no direct mention of differences between warm versus cold sites, although differences seem to exist (Fig. 2d, 2f).

[Response #R1.2] We thank the reviewer for pointing this important issue. Following the reviewer's suggestion, we explored the spatial difference in leaf unfolding and GDD between warm and cold sites. We found that, although background temperature appears to play a significant role in explaining leaf unfolding and GDD spatial variance, we did not find any difference in temperature effect between cold or warm sites (We edited the result section L. 202-204). One of the reason might be that the temperature range spanned by all sites is too small to clearly separate warm and cold sites (See Supplementary Figure 2). Moreover the temperature information is already partially captured by the interplay between chilling and forcing requirement at each site, and indeed we found that GDD and NCD are correlated (Supp. Figure 5, see Response #R3.2).

However, thanks to the reviewer's suggestion, we highlighted a key result regarding the effect of light, and more specifically incoming shortwave radiation, on the chilling-forcing relationship. We now show that the relationship between chilling and forcing requirement for leaf unfolding is shifted between sites with low and high pre-season radiation availability (Figure #R1.2). Here the relationship appears clearly at the spatial level when considering long-term averaged observations and incoming

radiations. We argue that this relationship is mainly an artifact induced by the fact that we used the same GDD and NCD definition for all sites.

This result strengthens one of our conclusions that commonly used GDD (and NCD) models are not suitable for large scale prediction if using a constant parameterization. GDD and NCD have been shown to be good predictors of the temporal variability of leaf unfolding at one site. However, because trees can have different sensitivity to temperature in different regions, it implies that different GDD and NCD definition have to be used at the spatial level for different sites. This is also the reason why GDD is not only determined by spring temperatures and NCD in our study, but is strongly correlated to other biogeographical conditions.

This result has large implications, since it challenges previous studies assessing GDD variability that have been considering a rigid GDD definition.

We edited the discussion section (L.341-349) to highlight this new result.

Figure #R1.2: Relationships between long-term averaged (1970-2016) forcing (defined as the growing degree days -GDD- with a threshold temperature of 5°C) and chilling requirement (defined as the number of chilling days -NCD- with a temperature between 0 and 5°C).

[Comment #R1.3] Also, the authors spend little time discussing species differences, despite some interesting patterns emerging from their analysis (Fig. 2b, 2d, 3e).

[Response #R1.3] We have now added a new section (L.284-307, "Species show different responses to biogeographical constraints") to discuss species differences in leaf unfolding response to climate following the insightful suggestion of the referee.

Introduction:

[Comment #R1.4] Line 51-52: "Studies have mostly focused on the temporal (interannual) changes in spring phenology, but..." Adding more specific detail here could help the reader to better understand the different focus of your study, i.e. 'Many studies have documented that recent warming trends have advanced leaf unfolding dates for many temperate tree species, but ...'

[Response #R1.4] We edited Lines 51-52 following the reviewer suggestion: “Several studies have shown that recent warming trends have advanced leaf unfolding dates for many temperate tree species, but the observed spatial heterogeneity in phenological LU trends at the global scale^{10,11} suggests a strong effect of water availability and light on the sensitivity of vegetation to global change.”.

[Comment #R1.5] Line 59: “mitigates”

[Response #R1.5] We corrected mitigates L.59.

[Comment #R1.6] Lines 62-64: The term “acclimation” has a very specific meaning to plant ecophysiologicalists, which may be different than your meaning here. I consider an acclimation response as occurring over days to months, allowing organisms to maintain homeostasis in response to environmental change, whereas adaptation requires decades to centuries (see Figure 1 in Smith and Dukes, 2013, Plant respiration and photosynthesis in global-scale models: incorporating acclimation to temperature and CO₂, Global Change Biology). Basically, are you meaning temporal variance at an interannual time scale? If so, within how many years? Clarification may require additional discussion of how quickly species can adapt their phenology to persistent warming trends.

[Response #R1.6] We agree with the reviewer's comment and changed the terms “adaptation” and “acclimation” into “response” to avoid any confusion. Following the reviewer’s comment, the discussion regarding the phenological response to persistent warming was added in the revised manuscript: “Previous studies showed a reduced sensitivity of leaf unfolding to climate warming in the last decades mainly attributed to plant plasticity⁵³. However, how trees acclimate or adapt to future climate change remains unclear and might be species dependent due to differences in temperature, light and water sensitivities.” (lines 296-300).

[Comment #R1.7] Lines 64-70: This example does a good job of highlighting the complexity of environmental differences between high and low elevations, i.e. potential factors affecting spatial variation in phenology, but it doesn’t help to clarify your theory for different mechanisms controlling temporal versus spatial variation. Could you expand this to better explain why temporal variation in spring phenology would be a short-term acclimation response? Rather, I have always interpreted temporal variation in spring phenology as an aspect of long-term adaptation to a temperate climate. For example, isn’t delayed budburst in a cold year (relative to a warm year) an adaptation to avoid frost damage?

[Response #R1.7] We agree with the reviewer, so we removed the example. The example provided L.55-56 is enough to highlight the existence of spatial differences in phenology.” At the regional scale, temperature differences have been proposed as the main cause of the spatial differences in mean LU date between high and mid-latitudes^{12,13}, low and high elevations¹⁴⁻¹⁶ or coastal and inland areas^{17,18} ”.

[Comment #R1.8] Line 80-81: Clarify, is NCD where the maximum or mean daily temperature is 0-5 C? (Actually, your methods indicate that it’s mean, but it would be good to indicate here too.)

[Response #R1.8] Following the reviewer’s suggestion, the sentence was clarified as: “NCD, defined as the number of days with a mean temperature between 0 and 5 °C from 1 November to LU” (lines 88-89).

[**Comment #R1.9**] Line 89-90: It would be helpful to include several other details here: Are the tree species all deciduous? What is your definition of “long-term”, i.e. what dates? What geographic spatial extent did you analyse? How many sites were included in your analysis?

[**Response #R1.9**] We edited the paragraph and included information as suggested by the reviewer: “8 dominant European deciduous tree species (see Methods) and thus to understand how LU and GDD* relate to long-term differences in temperature, radiation and aridity. We combined long-term (over 1970-2016) observations of LU dates for these eight tree species over Europe (27790 sites; Supplementary Figure 1) (L.112-115)”.

[**Comment #R1.10**] Line 95: day of year, DOY

[**Response #R1.10**] We edited the term accordingly in the revised manuscript (line 128).

Methods:

[**Comment #R1.11**] Line 314-318: A warming trend did exist over your study period (1970-2016), but given that you are using the median LU over this nearly 50-year period, this seems logical reasoning. I agree that it is appropriate to exclude responses from extreme years, but why didn't you also exclude extreme years when calculating mean climate variables?

[**Response #R1.11**] We thank the reviewer for this comment, and following the reviewer's suggestion, we redid the whole analysis by excluding also the extreme climatic years. The new methodology did not change the key messages of the analysis. We edited the corresponding method section in the revised manuscript: "As for LU, extreme climatic years were excluded as we seek to estimate the average response of the vegetation "(L. 429).

[**Comment #R1.12**] Lines 324-338: It seems redundant to have preseason (TP) and growing season temperature (TG) in the analysis (also GDD), for several reasons. There is likely a strong correlation between these variables across study sites. **Further, growing season temperature occurs after leaf unfolding, by definition, and thus is not a likely candidate for a phenological cue.** Also, inclusion of many variables makes interpretation more difficult. The same argument could be used for other variables, including radiation and precipitation. While the nature of your study is not necessarily focused on environmental cues for budburst phenology, but rather evolutionary pressures on leaf phenology, this important distinction was not made clear in the introduction.

[**Response #R1.12**] We agree with the reviewer regarding the nature of our study focusing on evolutionary pressures on leaf phenology. However, we argue that tree seasonality affects eco-physiological processes and is a strong driver impacting plant development. These aspects are still overlooked in phenological studies (and eco-physiological studies in general). The pressure on trees does not only occur during spring, but during the whole year, and there are increasing evidences that seasonality plays a key role in tree phenology, from the bud formation to leave senescence (see #R1.1 for a list of references, We edited L.66-75). We argue that confronting information from the preseason and the growing season are more representative of environmental pressures faced by trees than only using spring meteorological conditions. (We edited L.120-122).

As suggested by the three reviewers, we now checked for collinearity issues between GDD, TP and TG (See Response #R.1.1). We agree that increasing the number of predictors increases the complexity of result interpretation. Following the reviewer's suggestion, we now selected relevant predictors using penalized regressions and a “stepVIF” selection of variables to deduce a reduced model from all variables. Importantly, even after taking care of collinearity by different methods, the

results did not change, i.e. we observed that long-term site temperature (including TP and TG) is still selected as a key predictor of leaf unfolding. shows that commonly used GDD-NCD models are incomplete to fully capture the spatial variability of leaf unfolding (see Figure #R1.12).

Figure #R1.12 | Relative importance of each variable (in percentage) in explaining leaf unfolding date (LU) and corresponding heat requirement (GDD*, estimated as the sum of temperatures >5 °C between 1 January and the LU date) for each species considering a-c) all sites or b-d) only drought prone sites. The sum of each row is equal to 100%. Red colors indicate a positive correlation, while blue colors indicate a negative correlation. Color intensity reflects variable relative importance. Blanks represent variables that were discarded during the predictors selection. See caption of Figure 1 for the description of each variable. The direction and importance of correlations among species is summarized by the size and color of arrows at the bottom of each panel. A double black arrow means that the direction of the response is species dependent. Refers to Supplementary Tables 5-10 in appendix for a complete description of correlation coefficients.

We edited the whole results section with the new analysis (L.125-204).

We also edited the introduction section to clearly emphasize the focus of our study and the key hypothesis regarding the effect of biogeography and seasonality on tree phenology (L. 67-75 &101-109).

Results:

[**Comment #R1.13**] Your overall structure here does not do the best possible job of highlighting your main results, plus you return to explaining temperature effects under the “Precipitation” heading. Rather than using “Temperature”, “Radiation”, and “Precipitation”, you could split between “Wet temperate forest” and “Temperate forest with more frequent drought”. Were there any differences between warm versus cold sites?

[Response #R1.13] We edited and restructured the whole result section as:

- "Importance of temperature, light and water availability in capturing leaf unfolding spatial variability" (line125).
- " Importance of temperature, light and water availability in capturing heat requirement spatial variability" (line158).
- "Different response of leaf phenology to background conditions in drought-prone sites" (L.176)

See #R.1.2 for the response about warm/cold sites.

[Comment #R1.14] Figure captions: How many sites are included in analysis? It is okay to indicate a range of numbers to account for differences among species, but this is important information to describe.

[Response #R1.14] We now included the number of sites for each species in Table 1.

[Comment #R1.15] Figures 2-4: Please add in a brief description in the caption of what negative and positive numbers indicate. The interpretation is difficult for me and may be confusing for other readers too.

[Response #R1.15] Now Figure 2 to 4 were removed and replaced by the new analysis described in Response #R.1.1.

[Comment #R1.16] Lines 108-110: As mentioned previously, it is strange to make a distinction between GDD, TP, and TG. These are all temperature-based measures that are likely to be correlated across sites. Rather, I expected to read here about the contribution of radiation and precipitation.

[Response #R1.16] See Response #R1.12 for the rational of using TP and TG in our analysis.

[Comment #R1.17] Lines 115-118: This is interesting. Differences among species could be indicative of different adaptations that have resulted from evolutionary pressures by these environmental variables. I recommend adding more information to shift your results to this focus. For example, how much variation exists among species in leaf unfolding date and GDD, within a site? How does the range in LU/GDD across sites differ among species? Are all species found at all sites, or is the distribution of some species restricted to certain sites (i.e. wet sites)? Which species are considered more drought tolerant (e.g. Fagus and Quercus are generally considered to be more drought tolerant than Tilia and Sorbus) and does this help to explain results?

[Response #R1.17] Following the reviewer's suggestions, we improved the inter-species comparison of our results and a new discussion section was added as well in the revised manuscript (L.284-307). We also added more information about species and species distribution in Table 1 and Supplementary Figure 1.

[Comment #R1.18] Lines 158-159: In a global context, arid ecosystems are generally considered to have very little annual rainfall (<300 mm), e.g. deserts, drylands, etc. Therefore, I suggest changing your use of "arid" in table/figures to "drought-prone" or "drier" or "dry" to avoid confusion and fit your definition better. Or if you really wanted to examine only the driest sites, maybe you should consider using a lower AET:PET threshold than 0.9? For example, the largest differences among species in Figure 4d occur at AET:PET<0.8. Also, please state how many sites were characterised as "arid", at least in table/figure captions.

[Response #R1.18] Following the reviewer's suggestion, we changed "arid" by "drought-prone" sites throughout the manuscript. We kept the AET:PET threshold at 0.9 and indicated the number of drought-prone sites in Supplementary Table 1. Since it is an averaged value over 30 years, a threshold of 0.9 is already a good estimate for sites facing regular summer droughts. Moreover, we fixed this threshold based on the GDD variance evolution observed in Figure 5 (now Figure 3). Mathematically this threshold based on GDD iqr was estimated at 0.93 but we kept 0.9 to be sure that only drought-prone sites are selected in the analysis.

Discussion:

[Comment #R1.19] Lines 192-207, 236-244: Much of this section reads as if you are describing evidence for environmental cues of budburst phenology. Is this really the main focus of your analysis?

[Response #R1.19] These sections do not describe evidences for environmental cues of spring phenology but evidence for a significant constraint of leaf unfolding by biogeographical conditions, which is often overlooked in phenology studies. We edited the introduction section L.97 as well as the conclusion section L.330 to clarify this point.

[Comment #R1.20] Lines 210-213: But didn't your analysis pair responses to climate in the same year, not the previous year?

[Response #R1.20] Our study analyzes the biogeographical constraints on spring phenology, considering the average conditions faced by plants both during spring and during the growing season. (See Response #R.1.12 for the rational of including growing season conditions in our analysis).

[Comment #R1.21] Line 223: Cite relevant figures here, and throughout discussion.

[Response #R1.21] We now added references to relevant Figures throughout the discussion.

[Comment #R1.22] Lines 228-229: This hypothesis was not mentioned in the introduction. It would be helpful to set up these findings by describing potential hypotheses at the beginning. See also lines 205-206.

[Response #R1.22] We edited the introduction section to emphasize key hypothesis regarding biogeographical constraints and seasonality that we are addressing in our study. (L.65 & 97)

[Comment #R1.23] Lines 229-230: Clarify if you are referring here to intraspecific or interspecific variation. Are there any relevant species differences that could be described more specifically and referred to here?

[Response #R1.23] We added a new discussion section to deal with species differences (L.284-307).

[Comment #R1.24] Lines 247-248: But drought can also lead to early leaf senescence in some species, particularly those with low drought tolerance (see Marchin et al. 2010, Drought-deciduous behavior reduces nutrient losses from temperate deciduous trees under severe drought. *Oecologia*, 163: 845-854).

[Response #R1.24] Thanks for the reference. We also discussed that the drought plays a key role in the senescence processes, and cited the study by Marchin et al. (2010) in the revised manuscript (L.323)

Conclusion:

[Comment #R1.25] Lines 271-272: Isn't conclusion #2 only applicable to dry sites? Table 1 indicates that both radiation and daylength were only included in the GDD model for dry sites. Please clarify to prevent overstating your findings.

[Response #R1.25] We re-evaluated results based on the new methodology, and tested that this conclusion applied for all sites (see Figure #R1.12).

I hope that you find these comments helpful,

Renée M. Marchin

Western Sydney University

[Response #R1.26] Many thanks Dr. Marchin. Yes, your comments were very useful to improve the manuscript.

Reviewer #2 (Remarks to the Author):

General comments:

[Comment #R2.1] Marc Peaucelle and colleagues combine a large-scale European dataset of spring phenology from eight tree species with several climate variables to assess biogeographical effects of temperature, light and aridity on the timing of leaf unfolding (LU) and on growing degree days (GDD). For both LU and GDD, the long-term median from 1970 to 2016 was used at each site, for the climate variables, long-term means from 1970 to 2016 were used. While the spatial extent and the large amount of the data is truly impressive, I think, the study lacks scientific rigour to be published at the current stage. In want to briefly summarize the major concerns that I encountered:

[Response #R2.1] We thank the reviewer for all the very helpful comments that clearly improved the methodology, as well as the clarity and the robustness of our analysis.

[Comment #R2.2] While the study has been framed into a spatial context, there is apart from Supplementary Fig. 2 no spatially explicit analysis of the data. Rather than treating the spatial observations as independent, which was assumed for the linear models and the partial correlations, spatial autocorrelation needs to be accounted for. Otherwise, an important assumption of the linear model is violated. Treating spatially autocorrelated observations as independent results in biased standard errors, P values and R^2 values, thus significant effects are found more often than there actually are. There are many ways how to deal with spatial data (see e.g. Dormann et al. 2007). The “nlme” package in R provides several options to model spatial correlation structures (see Pinheiro and Bates 2000).

[Response #R2.2] We thanks the reviewer for these constructive comments. Following the reviewer’s suggestions, we performed additional analysis in order to evaluate the effect of spatial auto-correlation on our results:

1) After checking for co-linearity between variables (see Response #R1.1) we assessed the presence and the structure of the spatial auto-correlation using semi-variograms for each species and with all data pooled together (Supplementary Figures 6,7 &8).

2) Using the nlme package as suggested by the reviewer, we then performed new regressions by taking into account spatial auto-correlation using the gls function. We tested different spatial structures (linear, exponential, logistic and spherical, Supplementary Table 4) based on semi-variograms of residuals. The new results did not change our key messages. We edited the main text with the new analysis, both the Methods, Results and Discussion section (L.125-204;284-307; 335-360; 422-454).

[Comment #R2.3]The multiple linear models to predict LU date and GDD (Table 1) include not only long-term mean climatic variables from the pre-season (i.e. three-month period before LU) but also from the growing season (i.e. period from days 180 to 250). I was puzzled to see that climatic variables such as the mean growing-season temperature (TG), the mean growing-season short- and longwave radiation (SWG, LWG), the growing-season total precipitation (PG) and the growing-season soil-moisture content (SMG) were used to predict a phenological event (LU) that actually occurs at the beginning of the growing season (i.e. either prior to the observation period of these growing-season variables, or only in spring of the following year). While the authors tried to interpret these effects in the Discussion (L235-253), i.e. how the formation of buds during summer and their sensitivity to weather variability affects spring phenology, the current knowledge of phenological processes does not support these

[Response #R2.3] Assessing the constraint of biogeography on leaf unfolding is the core of our study. As we argued in Response #R1.1 and #R1.12, the growing season conditions also represent biogeographical constraints faced by trees, and not only spring meteorological conditions as it is always suggested. We fully agree with the reviewer that the key hypothesis addressed in our study were not clearly introduced, but the choice of using growing season climatic variables was made on purpose and not randomly integrated in the analysis. We edited the introduction in order to emphasize the focus and the key hypothesis of our study (L. 65, 97 & 101).

However, we cannot agree with the reviewer that, “the current knowledge of phenological processes does not support our hypothesis”. Firstly, because links between leaf phenology and plant activity during the growing season have already been highlighted (see Stromme et al., 2015 and Liu et al., 2018gcb for example), and secondly because there are recent evidences suggesting that tree seasonality significantly impacts eco-physiological processes that are inter-connected (Le dantec et al., 2000, Breda et al. 2006; Hänninen & Tanino, 2011, Fu et al., 2014; Keenan and Richardson 2015; Stromme et al., 2015; Liu *et al.*, 2018), we therefore explored the impact of growing season climatic cues on leaf unfolding and GDD . We edited the introduction section L.67-75 to highlight this point.

We argue that biogeography and seasonality are two overlooked components of tree phenology, and this is specifically what we tried to highlight with our study, in which we show that these two aspects play a significant role in spring phenology spatial variability.

We fully agree with the reviewer that the hypothesis developed here will need further investigation, and especially control experiments, to be evaluated temporally. The corresponding discussion and conclusion sections were therefore further improved in the revised manuscript (L.250-258; 314-318; 330-333; 380-383) too.

Hänninen, H. & Tanino, K. Tree seasonality in a warming climate. *Trends Plant Sci.* **16**, 412–416 (2011).

Le Dantec, V., Dufrêne, E. & Saugier, B. Interannual and spatial variation in maximum leaf area index of temperate deciduous stands. *For. Ecol. Manag.* **134**, 71–81 (2000).

Strømme, C. *et al.* UV-B and temperature enhancement affect spring and autumn phenology in *Populus tremula*. *Plant Cell Environ.* **38**, 867–877 (2015).

Bréda, N., Huc, R., Granier, A. & Dreyer, E. Temperate forest trees and stands under severe drought: a review of ecophysiological responses, adaptation processes and long-term consequences. *Ann. For. Sci.* **63**, 625–644 (2006).

Fu, Y. S. *et al.* Variation in leaf flushing date influences autumnal senescence and next year’s flushing date in two temperate tree species. *Proc. Natl. Acad. Sci.* 201321727 (2014).

Keenan, T. F. & Richardson, A. D. The timing of autumn senescence is affected by the timing of spring phenology: implications for predictive models. *Glob. Change Biol.* **21**, 2634–2641 (2015).

Liu, G., Chen, X., Zhang, Q., Lang, W., & Delpierre, N. (2018). Antagonistic effects of growing season and autumn temperatures on the timing of leaf coloration in winter deciduous trees. *Global change biology*, 24(8), 3537-3545.

[**Comment #R2.4**] results. On the one hand, logic reasoning rejects the possibility of retrospective effects of growing-season climate variables on spring phenology in the current year. On the other hand, **if the climatic conditions during the current growing season** (i.e. mainly July and August) **had such a strong impact on spring phenology of the following year** (i.e. more than 6 months following the predefined growing season), **this would be known for a long time**. Common knowledge, however, suggests that the onset of leaf unfolding is mainly driven by forcing temperatures in spring that interact with chilling temperatures in winter.

[**Response #R2.4**] We fully agree that a future event cannot affect the present process. Please notice that this was claimed in our study. Here, we highlighted that biogeography constrains leaf phenology on the long-term, and it appears that mean growing season conditions are a good proxy of that biogeographical constraint. Please also note that we are not looking at the temporality of spring leaf unfolding, but at the long-term effect of climatic conditions in order to explain spatial differences in phenology. We edited the introduction (L.65, 97) and discussion sections to better clarify why we considered growing season variables in our study (See also Response #R.1.1; #R1.12; #R.2.3).

However, we disagree with the reviewer saying “ if the climatic conditions during the current growing season (i.e. mainly July and August) had such a strong impact on spring phenology of the following year (i.e. more than 6 months following the predefined growing season), this would be known for a long time.”. First because our results do not show that growing-season conditions has a strong impact on spring phenology but rather that it significantly explains **spatial differences** in average leaf unfolding date at the spatial level. During the growing season, trees replenish reserves that will be used during the next leaf unfolding event and create new buds that will burst the following year. It is thus reasonable to hypothesize that growing-season conditions will constrain tree functioning on the long-term (We edited L.316 to emphasize this argument). This has been highlighted in Fu et al. 2014 but also in Strømme et al. 2015 (see the list of references about tree seasonality in #R2.3).

Secondly, because the argument that “this would be known for a long time” is not a scientific argument and is irrelevant in our context since it has never been really addressed before in the literature.

[**Comment #R2.5**] The main issue with these analysis is that correlation does not imply causality. Many of the effects that are shown in Table 1 or Fig. 1 occur because of the high correlation among variables. While at a warm site not only high spring temperatures are observed resulting in relatively early budburst, but also high summer temperatures, whereas at a cool site spring temperatures are rather low that result in delayed budburst, followed by cool summer temperatures. To detect potentially lagged effects of growing-season climate variables on leaf-out time or GDD in the following spring, site-specific models need to be used. Thus, instead of using long-term mean climate variables, the effects of climate variables from the current growing period on LU date or GDD of the following year need to be estimated.

[**Response #R2.5**] We thank the reviewer for the suggestions. We agree with the reviewer and we would like to answer to this comment in two points:

1) We revised our methodology (L.422-454) to assess potential collinearity issues induced by the use of multiple dependent variables (see Response #R1.1). The new analysis lead to the same conclusions regarding the significant role of biogeography in constraining LU (and GDD) at the spatial level. We added the correlation matrix between variable in Supplementary Figure 5 (Figure #R2.5).

2) Please notice that we are not looking at the temporality of processes but rather at the biogeographical constraints on LU. Thus, we are not looking at the effect of the previous year meteorological conditions on the current year spring phenology. As we described in previous responses, we argue that trees are not only affected by spring meteorology/ On the contrary, we argue that background growing season conditions are a better proxy of biogeographical pressure on trees than only spring meteorological conditions (see also Response #R.2.6).

Figure #R2.5 | Pearson correlation coefficient (in %) between predictors. GDD, growing degree day estimated as the sum of daily temperature above 5°C from the 1 January to leaf unfolding (LU). NCD, number of chilling days estimated as the number of days between 1 November in the previous year and the LU date with temperatures between 0 and 5 °C; TG, mean growing-season temperature; TP, mean pre-season temperature; SWG, mean growing-season shortwave [visible and near infrared] radiation; LWG, mean growing-season longwave [infrared] radiation; SWP, mean pre-season shortwave [visible and near infrared] radiation; LWP, mean pre-season longwave [infrared] radiation; PG growing-season total precipitation; PP, pre-season total precipitation; DL, day length at LU date; SMP, pre-season soil-moisture content; SMG, growing season soil-moisture content; αE, ratio of actual to potential evapotranspiration.

[Comment #R2.6] I was further surprised to see (1) that temperature and precipitation during the growing season are suggested to have almost the same relative importance as GDD, and (2) that the number of chilling days (NCD) has actually the highest relative importance for explaining LU date (see Fig. 1a). For most species NCD seems to be more important than GDD for predicting LU, and for all species the mean growing-season temperature (TG) is more important than GDD (Supplementary Fig. 4). If these findings were true, they would question most phenological studies that have been previously published. However, a majority of mechanistic phenology models does not even show any significant contribution of chilling on leaf-out time (not to mention of growing-season variables of the previous year), i.e. the main driver seems to be forcing in spring (e.g. GDD; see e.g. Basler 2016).

[Response #R2.6] As said, please notice that we are not looking at the temporality of LU but at its spatial variance, and substantial underlying mechanisms might exist between these two variances in LU.

With the new methodology (please see the updates in L.422-454 in the revised manuscript), we found that NCD as the same weight than GDD in explaining spatial variance of LU (see Figure 2). We would like to address reviewer's comment from two aspects:

First, for the same observation dataset, chilling has been shown to play on the temporality of phenology (see our previous results in Fu et al. 2015). Moreover, in several phenology papers like Basler 2016, the definition of forcing and chilling can differ a lot. Here we used the most basic, and commonly used definition of GDD and NCD, and we assumed that all species at all sites have the same sensitivity to both driver, which might be oversimplified but commonly used in phenology studies (including Basler 2016). Using a constant parameterization here translated in a high sensitivity of GDD to environmental clues. This relationship between GDD and site temperature has already been reported by Jenkins et al. (2002) and Maignan et al. (2008). Our results illustrate that current mechanistic models are not suitable for predicting spatial leaf unfolding variation, even if they perform relatively well in predicting temporal patterns. We think that our results will help the modeling community in building better process models by giving new insight regarding the spatial, long-term variability of leaf unfolding.

We edited the abstract (L.38), the introduction (L.104), the discussion (L.335) and the conclusion (L.381) to emphasize this point.

Secondly, TG was still selected as a significant predictor of the spatial variability of LU. Two hypothesis can explain this result: 1) TG is a good proxy of site biogeographical constraints on LU, and potentially summarizes other variables not considered in our study or 2) trees growing at different locations have optimized the control of their LU in response to biogeographical differences in pre-season and growing season conditions. In both cases, this result indicates that site biogeographical conditions have to be taken into account in addition to commonly used pre-season temperatures that do not suffice to explain the spatial distribution of LU. We edited L. 218 to highlight this point.

Jenkins, J., Braswell, B., Frohling, S. & Aber, J. Detecting and predicting spatial and interannual patterns of temperate forest springtime phenology in the eastern US. *Geophys. Res. Lett.* **29**, (2002).

Maignan, F., Bréon, F., Vermote, E., Ciais, P. & Viovy, N. Mild winter and spring 2007 over western Europe led to a widespread early vegetation onset. *Geophys. Res. Lett.* **35**, (2008).

Specific comments:

[**Comment #R2.7**]L64-70: The sentence should be split up into two sentences.

[**Response #R2.7**] According to Comment #R1.7 we have removed this paragraph.

[**Comment #R2.8**]L67, L121 etc.: Replace “altitude” with “elevation” (see McVicar and Körner 2013).

[**Response #R2.8**] We replaced altitude with elevation throughout the text.

[**Comment #R2.9**]L88-93: Forest trees are affected by local light conditions, which can strongly deviate from large-scale conditions. Thus, the local light regime is probably more important than then overall light regime (compare e.g. a free-standing tree at high latitude with a tree growing under a dense canopy at low latitude.).

[**Response #R2.9**] Thanks for rising this aspect. We now edited the discussion section L226 to discuss this important point: “However, differences in stand canopy openness, leaf area or even plant activity can lead to large uncertainties in the potential effect of growing season radiation on tree eco-physiology due to differences in local light regimes. As for TG, both SWG and LWG can only be good proxies of background biogeographical constraints without having a direct effect on phenology.” (L.254-258)

[**Comment #R2.10**]L89: I suggest to define "long-term" also in the main text, otherwise this only gets clear when reading the methods (L314).

[**Response #R2.10**] We now defined “long-term” in the introduction as :” long-term (over 1970-2016) observations of LU” (L. 113).

[**Comment #R2.11**]L95, L325-328: Days 180 to 250 probably reflect the growing period at just a few sites (high latitudes or high elevations). However, at many sites the growing period starts several month before and ends 1-2 months afterwards. E.g. the mean temperature calculated over the actual growing period can substantially differ from the mean temperature over days 180 to 250. Thus, mean values from day 180 to 250 do not generally represent conditions during the growing season.

[**Response #R2.11**] Indeed, trees have different growing season length depending on site and species, and the growing season length varies from one year to another. Since we are interested by biogeographical conditions and not the temporality of processes, selecting the summer period between days 180 and 250 ensures that we have a representative period for all trees and all years, and that we are able to compare long-term conditions of all sites without any bias induced by different growing season lengths. This period also usually corresponds to the “peak” of plant activity in temperate ecosystems, and this is the period over which we are more likely to take into account for water and temperature pressures of sites. We edited the method section to highlight this point (L.422-428).

[**Comment #R2.12**]L99: Unclear why the subtitle is “temperature”. Other variables not related to temperature are presented as well, e.g. radiation or water availability. These results should be moved to "Radiation and day length" and "Precipitation and aridity", respectively.

[**Response #R2.12**] With the new analyses based on the new methodology, we edited the whole result section as well as all subtitles.

[Comment #R2.13]L100: Stepwise regression should not be used because of various shortcomings (e.g. p. 56 in Harrell 2001), instead information-theoretic model selection should be used (e.g. Stauffer 2008).

[Response #R2.13] We revised the whole methodology as described in Response #R1.1 and we removed the stepwise AIC regression method from our analysis.

[Comment #R2.14]L104: The R^2 values are biased because of spatial autocorrelation (see e.g. Dormann 2007 or Lennon 2000).

[Response #R2.14] See the detailed Response #R2.2 about spatial auto-correlation.

[Comment #R2.15]L109: What is the causality between growing season temperature (TG) and LU? If growing season variables show an effect, this is because of correlations with preseason variables (e.g. at a warm site, temperature is high during the preseason as well as during the growing season).

[Response #R2.15] By playing on senescence and thus by reducing the dormancy length and chilling accumulation, TG can have an indirect impact on subsequent leaf unfolding (see Fu et al. 2014)

Please also see previous responses about the rationale for using TG.

[Comment #R2.16]L113: Standard errors need to be provided for all slope estimates.

[Response #R2.16] We now provided the standard error for regression slopes in the revised manuscript (Supplementary Tables 2-7).

[Comment #R2.17]L125-127, L146-147, L156-157: Here, too, the question of causality arises.

[Response #R2.17] We revised the methodology to deal with collinearity issues (see Response #R1.1 and #R2.5).

[Comment #R2.18]L129, L145-151 etc.: I recommend to provide exact P values (not just significance levels).

[Response #R2.18] We now added exact p values in Supplementary tables 7-10.

[Comment #R2.19]L131-132: However, the effect of SWP and LWP on LU was not tested, because the variables were not included in the models (see Table 1, L554-555).

[Response #R2.19] Daylength and incoming radiation increase each day from the 1st January until the summer solstice, thus a later leaf unfolding translates into a longer daylength and higher preseason incoming radiation. The correlation between LU and DL or SWP is > 0.9 . This is the reason why we removed SWP, LWP and DL from the LU model. We included the correlation matrix in Supplementary Figure 5 and we edited the method section to explain this point (L.432). We also edited the discussion section L.226-262.

[Comment #R2.20]L144-145: However, the effect seems to be significant ($p < 0.05$). Just because the estimated coefficient is small does not mean that LU date is not sensitive to PG. The coefficient depends on the scale on which PG is measured.

[Response #R2.20] This is right. We now added exact P values in supplementary tables and we reformulated the main text where necessary.

[**Comment #R2.21**]L183: However, in temperate forests, forcing (e.g. GDD) is generally more important than chilling, because chilling conditions are fulfilled related to the rather cool winter conditions.

[**Response #R2.21**] We agree that GDD is more important than chilling when looking at the temporality of LU at sites. However, we are looking here at the biogeographical determinant of LU spatial variance, which might explain the observed difference compared to classical approaches. Due to the new method applied, we re-edited the whole result section.

[**Comment #R2.22**]L184-185: How can this effect be explained? TG represents the mean growing season temperature (e.g. a variable that is measured after observing LU).

[**Response #R2.22**] See Response #R2.6 and #2.15.

By playing on senescence and thus by reducing the dormancy length and chilling accumulation, TG can have an indirect impact on subsequent leaf unfolding (see Fu et al. 2014)

[**Comment #R2.23**]L210-213: If this was the case, the site-specific effects of radiation need to be quantified. Thus, the effect of radiation during the current growing season on LU date and GDD in the following year needs to be assessed. The current models do not allow to draw these conclusions because long-term variables are used.

[**Response #R2.23**] Following the reviewer's suggestion, we reformulated the discussion section as: "The growing season is also the period during which buds are created. In the long-term, we expect that trees respond to growing season meteorological conditions during which the formation of buds, but also carbon reserves, can be affected, which in turn affects the sensitivity of buds to temperature in the subsequent winter and spring as already shown with *Populus tremula*²⁴. However, differences in stand canopy openness, leaf area or even plant activity can lead to large uncertainties in the potential effect of growing season radiation on tree eco-physiology due to differences in local light regimes. As for TG, both SWG and LWG can only be good proxies of background biogeographical constraints without having a direct effect on phenology.

We argue that incoming radiation should be included in a phenological study, not only day length as a proxy of photoperiod.", L.250-258.

[**Comment #R2.24**]L242-244: I guess this is a purely correlative effect: at a warm site, temperature is not only higher during the growing season but also during the pre-season.

[**Response #R2.24**] See previous responses #R2.6 and #R2.15 related to the same comment.

[**Comment #R2.25**]L314: However, the date of leaf unfolding may differ by 1-2 months at a specific site, i.e. there is large variability that is not accounted for with the median.

[**Response #R2.25**] Since we are interested here in the spatial variability of LU and not the temporal variability, using the median values by sites is more relevant than using mean or extremes values to assess long term biological constraints on spring phenology. Moreover, we excluded extreme years from the analysis. We added Table 1 showing the intra and inter-site variability by species.

[**Comment #R2.26**]L335: Write "collinearity".

[**Response #R2.26**] We corrected collinearity throughout the text.

[**Comment #R2.27**]Figure 1: (1) Why is NCD more important than GDD? If most observations were from warm sites where chilling requirements are often not fulfilled, I might be inclined to believe this.

However, many sites are from areas with a clearly seasonal climate (i.e. alternation between rather warm summer and cool winter).

[Response #R2.27] See response #R.2.21 related to the same question.

[Comment #R2.28](2) E.g. the relative importance of TG on GDD is of purely correlative nature (i.e. both mean growing-season temperature and GDD at warm sites are higher than at cool sites).

[Response #R2.28] We performed the new analysis by taking care of collinearity issues and spatial auto-correlation. TG is still selected as a significant predictor of GDD. See Response #R2.6 related to the same question.

[Comment #R2.29]Figure 2: Write “residuals” (y axis label).

[Response #R2.29] We removed Figure 2 from the manuscript.

[Comment #R2.30]Figure 5: It seems that there is an offset, i.e. the black line should be shifted down to actually represent the trends for all sites and species.

[Response #R2.30] Thanks for rising this point. There is no offset, the black line represent the interquartile (iqr) calculated with data from all species pooled together. Since different species have different median LU date, the iqr with all species is higher than the iqr by species.

[Comment #R2.31]Table 1: (1) It should be explained why some variables were not included in the model (L554-555 and Supplementary Figure 3).

[Response #R2.31] We generated a new Table with the revised methodology. We took care of explaining the variable selection process in the method section (L.432-454). See also Response #R.2.19.

[Comment #R2.32](2) Standard errors / confidence intervals and exact P values need to be provided. Just based on the coefficients, it does not get clear, which variables are important and what the uncertainty of the estimates is.

[Response #R2.32] We now reported standard errors and exact P values in Supplementary tables.

[Comment #R2.33](3) LU and GDD are expressed by TG, SWG, LWG, PG, and SMG, i.e. values that are measured after LU or more than 6 months before LU in the following spring (see my previous comments about causality).

[Response #R2.33] See previous responses related to the same issue.

[Comment #R2.34](4) The spatial autocorrelation of the data is likely to result in biased R2 values.

[Response #R2.34] See previous responses related to the same issue.

[Comment #R2.35]Supplementary Figure 4: The variables SWP, LWP and DL are mentioned in the legend, but are not shown in the figure. A brief explanation for GDD is missing in the legend. R2 values are not shown in the figure.

[Response #R2.35] We now removed Supp. Figure 4 from the manuscript since we revised the whole methodology.

[Comment #R2.36]Supplementary Figure 5: The variables TP and DL are not shown in the figure. R2 values are not shown in the figure.

[Response #R2.36] We now removed Supp. Figure 5 from the manuscript since we revised the whole methodology.

[Comment #R2.37]Supplementary Figure 7: R2 values are not shown in the figure.

[Response #R2.37] We now removed Supp Figure 7 from the manuscript since we revised the whole methodology.

References:

- Basler D. 2016. Evaluating phenological models for the prediction of leaf-out dates in six temperate tree species across central Europe. *Agricultural and Forest Meteorology* 217:10-21.
- Dormann C.F. 2007. Effects of incorporating spatial autocorrelation into the analysis of species distribution data. *Global Ecology and Biogeography* 16:129-138.
- Dormann C.F., McPherson J.M., Araújo M.B., Bivand R., Bolliger J., Carl G., Davies R.G., Hirzel A., Jetz W., Kissling W.D., Kühn I., Ohlemüller R., Peres-Neto P.R., Reineking B., Schröder B., Schurr F.M. and Wilson R. 2007. Methods to account for spatial autocorrelation in the analysis of species distributional data: a review. *Ecography* 30:609-628.
- Harrell F.E. 2001. *Regression modeling strategies*. Springer Verlag, New York, Berlin, Heidelberg.
- Lennon J.J. 2000. Red-shifts and red herrings in geographical ecology. *Ecography* 23:101-113.
- McVicar T.R. and Körner C. 2013. On the use of elevation, altitude, and height in the ecological and climatological literature. *Oecologia* 171:335-337.
- Pinheiro J.C. and Bates D.M. 2000. *Mixed-effects models in S and S-PLUS*. Springer, New York.
- Stauffer H.B. 2008. *Contemporary Bayesian and frequentist statistical research methods for natural resource scientists*. Wiley and Sons, Hoboken, New Jersey.

Reviewer #3 (Remarks to the Author):

[Comment #R3.1] The authors present a study using the PEP phenology database to analyze spatial climate controls on phenological sensitivity to a variety of important variables (e.g., GDDs, light, chilling etc). Overall this is an interesting and detailed study, but I have some significant methodological concerns that I feel need to be addressed.

[Response #R3.1] We thank the reviewer for the positive feedback on our study. We revised the whole methodology following the reviewer's comments as detailed in Response #R1.1 and revised the methods section L.432-454.

[Comment #R3.2] The first, and most important in my mind, is the multiple stepwise linear regression results (Table 1 and Figure 1). Specifically, there is significant collinearity across all of the predictor variables that to me suggest these models are way overfit. For example, sites with warmer growing and pre-season temperatures are likely to also be sites with greater growing degree days and lower chilling. But it's not even just the temperature variables alone. Because both temperature and solar radiation both typically decline with latitude, these variables are also likely to be significantly correlated. Dry sites as well are also typically warmer in this region of the world, and so again moisture and temperature will not be independent. Nowhere, however, can I find any evidence that the authors have addressed these issues and, indeed, I'm not even sure it is possible given the largely inseparable interactions between these variables. Maybe this is dealt with in the partial correlation analysis, but it's unclear how this translates to the multiple regression, which by all accounts does not address collinearity issues.

[Response #R3.2] We agree with reviewer's comment regarding collinearity issues between climatic variables. Collinearity cannot be totally removed (Dormann et al. 2013), but we now followed recommendations to deal with collinearity as suggested in Dormann et al. 2013. We assessed collinearity between variables with correlations as recommended in Dormann et al. 2013, variance inflation factor (VIF) and penalized regressions. We then selected significant predictors in two steps, 1) with penalized regressions (elastic net) in combination to the correlation information, followed by VIF selection (with $VIF < 4$) of relevant predictors.

As described in the responses for reviewers #1 and #2 we also corrected coefficients for spatial auto-correlation. We edited the whole manuscript with the new method, results and discussion.

Dormann, C. F., Elith, J., Bacher, S., Buchmann, C., Carl, G., Carré, G., ... & Münkemüller, T. (2013). Collinearity: a review of methods to deal with it and a simulation study evaluating their performance. *Ecography*, 36(1), 27-46.

[Comment #R3.3] Second-I don't understand why the growing season predictors are even included, given that these metrics are dominated by things happening AFTER leaf unfolding. Why, then, should we even consider them in the first place? I guess the argument is that the impact from this season would manifest over evolutionary timescales, but it still feels like a bit of a weak argument without other evidence to back it up

[Response #R3.3] We explored the growing season predictors during previous years, not current year after leaf unfolding. As described in #R1.1, #R1.12 and #R2.3, one of the main hypothesis that we seek to explore with this study is that tree phenology is constrained, on the long term, by background biogeographical conditions and not only spring meteorology. We argue that confronting pre-season and growing season conditions is more representative of biogeographical constraints than using spring

conditions only. We thus decided to use multiple climatic drivers representing different phases of the life cycle of trees because recent evidences suggest that tree seasonality plays a key role on eco-physiological processes.

Since buds are formed the year before, during the growing season, we argue that any effect of growing season conditions does not necessarily takes evolutionary timescales. However, we fully agree that the significant effect of growing season variables observed here might only be a proxy of the biogeographical constraints on leaf unfolding.

To avoid confusion, we revised the introduction, discussion and methods sections (L. 65-75; 97-109; 250-258; 316-318; 350-360; 422-428) to highlight these points.

Reviewers' comments:

Reviewer #1 (Remarks to the Author):

This study investigated which environmental drivers influence the spatial variation of leaf unfolding in 8 temperate deciduous tree species in Europe. The median leaf unfolding date from 1970-2016 was assumed to represent long-term optimal leaf unfolding date for each site and was compared to potential (1) temperature, (2) radiation, and (3) precipitation/aridity drivers (using means from 1970-2016). The statistical analysis has been much improved from the previous version, and the authors have now accounted for spatial auto-correlation among sites and collinearity within the set of 13 climatic variables.

Temperature was the most important factor controlling spatial variance in leaf unfolding for all species, but radiation and precipitation/aridity also had significant effects. The discussion outlines several testable hypotheses for relationships between leaf phenology and radiation/water availability, providing direction for further research. It also does a good job of highlighting potential impacts on prediction of future changes in leaf phenology using regional models. This study is unique in that it is examining drivers of spatial differences in leaf phenology, rather than temporal differences, which are more commonly studied. Some minor changes are still needed to help clarify this important distinction, both in the Introduction and Discussion. Readers could easily misinterpret findings as being relevant environmental cues for budburst, rather than evolutionary pressures on budburst. The writing is sometimes vague so could be improved with more explicit word choice.

Abstract

Line 39: I think "matched to" may be a better word choice than "corrected for".

Introduction

There are several ways to improve the clarity:

Line 64: Please clarify, do you mean "different genetic mechanisms" or maybe "different mechanisms of control"?

Lines 67-71: This sentence is long and hard to understand; could you please clarify? Maybe separate into several concise sentences, e.g., "Biogeographical constraints on leaf unfolding include all environmental variables that may influence long-term adaptation to local climate, such as radiation

and drought. There are increasing clues suggesting that tree seasonality, defined as the annual cycle of growth and dormancy where one phase affects subsequent phases, also plays a key role...”

Lines 97-99: It is helpful to state your hypothesis here, but as currently stated, it reads more as a fact and less as a testable hypothesis. Can you please make this more specific? Perhaps your hypothesis is that growing season climate is more important than spring climate.

Line 101: “Compared to classical phenology studies (i.e. examination of environmental controls on temporal variation in leaf unfolding), ...”

Methods

Line 432: “pairwise” is misspelled

Line 442: “increase” not “increases”

Results

Figure 4: Is there a gradient colour scheme that does not have white? It is hard to see the light white points. This figure is also not yet described in the Results.

Table 1: The species seem to be in alphabetical order, except the last two species are switched.

Supplementary Figures 6-8: The axis titles and axis labels are very small and hard to read, could you please enlarge?

Lines 136-137: So is it 50% or 30%? Should it say, “accounted for less than half”?

Lines 166-167: Same as previous comment, I can’t decipher the difference between 52% and 28%... can you please clarify?

Lines 169-170: Interesting! Although this data doesn’t match perfectly with previous reports for photoperiod sensitivity among species, as *Alnus glutinosa*, *Fagus sylvatica* and *Tilia cordata* have had photoperiod sensitivity and *Sorbus aucuparia* was insensitive to photoperiod (see Way &

Montgomery 2014, Photoperiod constraints on tree phenology, performance and migration in a warming world, PCE).

Line 189-190: Two species names need to be italicised.

Discussion

Line 217-224: I find this paragraph confusing. As stated, it seems to be making the point that growing season temperature is important for leaf unfolding, without clarifying that it may be an important evolutionary pressure on leaf unfolding (not an important environmental cue for leaf unfolding). It will already be easy for readers to mistake your meaning, given that most studies analyse temporal and not spatial phenology patterns, so you will need to be very clear throughout to avoid confusion.

Line 236: “may be more important”? The title of this subsection states that radiation IS more important; which is correct?

Line 242-244: Could you elaborate on these two studies to help support your hypothesis? As currently stated, this hypothesis is rather vague.

Line 256: I think you mean, “... both SWG and LWG may only be ...”

Line 262: Delete “v”

Line 275-279: Interesting and reasonable hypothesis! It also provides additional time for rain events that may alleviate drought conditions.

Line 287: Figure reference here?

Line 288: Do you mean “high drought sensitivity”? Or perhaps “low drought tolerance” would be more clear.

Line 289: This would be more relevant: “*A. glutinosa* naturally occurs in wet sites”.

Line 291: Replace “drought tolerance capacity” with “low drought tolerance”.

Line 346-347: Interesting! To drive this point home, could you explain how the GDD definitions would differ? For example, “Northern sites would need to have lower (?) temperature thresholds for GDD than southern sites.”

Conclusion

Line 370: Change “control” to “influence”.

Line 382: Delete, “... and always accompanied by analysis of background environmental constraints and evolutionary pressures.” This will not be possible for many studies, due to logistics but also to a lack of knowledge for the specific mechanisms involved in these processes.

I hope that you find these comments helpful,

Renée M. Marchin

Western Sydney University

Reviewer #3 (Remarks to the Author):

The authors have addressed my concerns and I now recommend acceptance.

Reviewer #4 (Remarks to the Author):

I came into this after the revision, asked particularly for a comment on the statistics (collinearity).

I think the authors treated the issue of collinear predictors very well. The glmnet approach is IMHO one of the best ways to address collinearity for predictive models. I have no issue with this part of the analysis.

Reading the manuscript, I remain confused about several separate issues, which I detail here.

1. Line 78 defines LU as function of GDD, while the analysis uses GDD (and other variables) as predictors. I understand the point: process models define LU in this way, and the authors show that this is too simplistic. However, since the authors never give the actual regression “equation” (something like $LU \sim N(\mu = f(\text{GDD5}, \dots), \text{sd}=\text{const.})$ or so), the reader may easily be confused into taking the equation of line 78 as the definition of LU. I would scrap this equation, or at least relegate it out of sight into the supplement.

Similarly, L85 increases this confusion, as it is not clear whether the authors use LU as defined above, or as independently measured response variable. In the former case, the sentence is strange and should be “LU is a function of GDD and keeping ...”. In the latter case, the sentence should be something like “Observed LU might behave differently to the way it is represented in process models, as a function of GDD5.”

2. What is actually the RHS of the glmnet model?

Fig. 1 suggests that a range of predictors were used, but Table 1 shows that also their interactions were included. How then did the interaction feature in Fig. 1? Which variable got the partial r^2 of the interaction? Were also non-linear terms (i.e. quadratic) part of the model (or if not, why not: do the authors think that all effects are linear)? Was “species” a predictor in the glmnet, and if so, why not as a random effect in interaction with all predictors (as every species’ LU responds differently to the environment, it seems).

3. How comes that in Fig. 1 GDD is irrelevant (a and c), but in Fig. 2 it is among the darkest colours for several species, indicating its importance?

4. Fig. 3 a: This figure suggests a tiny effect of aridity on GDD until LU. Is this the basis for claiming that aridity modifies the effect of GDD on LU? If so, I would call the evidence flimsy at best.

Fig. 3 b: What’s the point of this figure? It seems to be neither referred to in the text, nor interpreted there.

Fig. 4: So, when it is on average warming in winter, then there are fewer chilly days? This blatantly obvious fact warrants a figure? And what is the switch from red to blue supposed to indicate, and where does the turning point of 125 Wm^{-2} come from? Again: what’s the point of this figure?

Overall, I found this MS highly confusing. It seems to make the important point that GDD is not a straightforward predictor of LU, as represented in various physiological models. Fine. The analysis of what actually drives LU could then be made much clearer, without dragging GDD along. Apparently, TG (mean growing season temperature) seems to be a very important predictor. However, as the other reviewers commented, how can it PREDICT LU, if it comes AFTER LU? The responses to the reviewers seem waffle to me.

Reviewers' comments:

Reviewer #1 (Remarks to the Author):

This study investigated which environmental drivers influence the spatial variation of leaf unfolding in 8 temperate deciduous tree species in Europe. The median leaf unfolding date from 1970-2016 was assumed to represent long-term optimal leaf unfolding date for each site and was compared to potential (1) temperature, (2) radiation, and (3) precipitation/aridity drivers (using means from 1970-2016). The statistical analysis has been much improved from the previous version, and the authors have now accounted for spatial auto-correlation among sites and collinearity within the set of 13 climatic variables.

[Comment #R1.1] Temperature was the most important factor controlling spatial variance in leaf unfolding for all species, but radiation and precipitation/aridity also had significant effects. The discussion outlines several testable hypotheses for relationships between leaf phenology and radiation/water availability, providing direction for further research. It also does a good job of highlighting potential impacts on prediction of future changes in leaf phenology using regional models. This study is unique in that it is examining drivers of spatial differences in leaf phenology, rather than temporal differences, which are more commonly studied. Some minor changes are still needed to help clarify this important distinction, both in the Introduction and Discussion. Readers could easily misinterpret findings as being relevant environmental cues for budburst, rather than evolutionary pressures on budburst. The writing is sometimes vague so could be improved with more explicit word choice.

[Response #R1.1] We thank the reviewer for the positive feedback on our manuscript and for providing insightful recommendations that clearly improved the main text. As suggested, we rephrased and reorganized the Introduction and Discussion sections to clearly emphasize the spatial aspect of our study (see responses #R1.2 - #R1.29).

Abstract

[Comment #R1.2] Line 39: I think “matched to” may be a better word choice than “corrected for”.

[Response #R1.2] We reformulated the abstract for more clarity. The sentence L. 39: “It suggests that common GDD models are not suitable for regional phenology studies if not corrected for background climate conditions.” was replaced by “This adaptation of GDD_{req} to background climate implies that models using constant temperature response are inherently inaccurate at local scale.” (L41).

Introduction

There are several ways to improve the clarity:

[Comment #R1.3] Line 64: Please clarify, do you mean “different genetic mechanisms” or maybe “different mechanisms of control”?

[Response #R1.3] We edited L. 74 as “different mechanisms of control”.

[Comment #R1.4] Lines 67-71: This sentence is long and hard to understand; could you please clarify? Maybe separate into several concise sentences, e.g., “Biogeographical constraints on leaf unfolding include all environmental variables that may influence long-term adaptation to local climate, such as radiation and drought. There are increasing clues suggesting that tree seasonality, defined as the annual cycle of growth and dormancy where one phase affects subsequent phases, also plays a key role...”

[Response #R1.4] We edited and simplified the whole paragraph to clearly emphasize the differences between temporal and spatial aspects of phenology (L. 74-88): “First, short-term,

fast responses to changing weather should drive temporal variations in LU and its GDD_{req} , aiming to maximize fitness under inter-annually varying weather conditions. Second, an adaptive response to local biogeographical conditions may maximize tree fitness under the local long-term mean climatic conditions and would select for spatially optimized LU and climate sensitivity, inducing spatial variation therein. Biogeographic constraints on LU include all environmental variables that impose long-term adaptation of LU and its GDD_{req} to optimize fitness under local conditions. These include climatic variables, such as site-specific occurrence of late frost events, drought occurrence, low or high light extremes, that may need to be avoided and therefore require shifts in growing season to enable maximum tree fitness. Also, site-specific interactions with neighboring competitors, pathogens and herbivores may induce spatial differences in LU and its weather dependency, in order to maximize tree fitness. Taken together, this suggests a complex response of plant phenology to climate change, but also that models of LU that apply spatially uniform parameters may not capture the observed patterns of LU and its GDD requirement.”

[Comment #R1.5] Lines 97-99: It is helpful to state your hypothesis here, but as currently stated, it reads more as a fact and less as a testable hypothesis. Can you please make this more specific? Perhaps your hypothesis is that growing season climate is more important than spring climate.

[Response #R1.5] We reformulated the hypothesis L.90-93 to be more specific: “The key hypothesis that we explore in this study is that long-term mean background biogeographical conditions determine the spatial heterogeneity of spring LU and its GDD_{req} , reflecting evolutionary mechanisms through which plants have adjusted their growth strategies in order to maximize their fitness under those specific biogeographical conditions.”

[Comment #R1.6] Line 101: “Compared to classical phenology studies (i.e. examination of environmental controls on temporal variation in leaf unfolding), ...”

[Response #R1.6] We edited and simplified the whole paragraph L.95-99 for more clarity and to emphasize the importance of growing season conditions: “We argue that biogeographic constraints on plant phenology can be detected by analyzing the spatial response of stands long-term mean LU and GDD_{req} , instead of their inter-annual variability. In this respect, we hypothesized that not only spring, but also mean growing season conditions are important controls of the spatial differences in leaf phenology among different locations. Altogether, these evolutionary mechanisms control the sensitivity of LU to short-term spring temperature variations, and consequently are key components of the observed spatial variability in LU and its GDD_{req} .”

Methods

[Comment #R1.7] Line 432: “pairwise” is misspelled

[Response #R1.7] We corrected the word “pairwise” L.430

[Comment #R1.8] Line 442: “increase” not “increases”

[Response #R1.8] We corrected “increases” by “increase” L.441

Results

[Comment #R1.9] Figure 4: Is there a gradient colour scheme that does not have white? It is hard to see the light white points. This figure is also not yet described in the Results.

[Response #R1.9] We redid Figure 4 and replaced the white color with a grey color. We also

updated the caption of Figure 4 (L. 663). Figure 4 was generated to support the Discussion and is described in the section L. 238-247.

[Comment #R1.10] Table 1: The species seem to be in alphabetical order, except the last two species are switched.

[Response #R1.10] We inverted the two last lines of Table 1 to have species in alphabetical order.

[Comment #R1.11] Supplementary Figures 6-8: The axis titles and axis labels are very small and hard to read, could you please enlarge?

[Response #R1.11] We enlarged axis titles in supplementary figures (now 7-9).

[Comment #R1.12] Lines 136-137: So is it 50% or 30%? Should it say, “accounted for less than half”?

[Response #R1.12] We agree with the reviewer that this sentence is confusing. In this sentence we made a distinction between the “explained” variance by the model and the “total” variance. The model explained $61 \pm 7\%$ of the total variance. Thus, half of the explained variance is $\sim 30\%$ of the total variance. We reformulated the whole sentence L. 130-132 to avoid this confusion: “Chilling and GDD_{req} together accounted for only half of LU’s explained variance (Figure 2a).”

[Comment #R1.13] Lines 166-167: Same as previous comment, I can’t decipher the difference between 52% and 28%... can you please clarify?

[Response #R1.13] Same as Response #R1.13. We reformulated the whole sentence L.160 to avoid the confusion: “together explaining $52.3 \pm 4\%$ of the spatial variance in GDD_{req} ”(Figure 2c).”

[Comment #R1.14] Lines 169-170: Interesting! Although this data doesn’t match perfectly with previous reports for photoperiod sensitivity among species, as *Alnus glutinosa*, *Fagus sylvatica* and *Tilia cordata* have had photoperiod sensitivity and *Sorbus aucuparia* was insensitive to photoperiod (see Way & Montgomery 2014, Photoperiod constraints on tree phenology, performance and migration in a warming world, PCE).

[Response #R1.14] Yes, a sensitivity to photoperiod has been reported for *Alnus glutinosa*, *Fagus sylvatica* and *Tilia cordata* in the review of Way and Montgomery. Most recent studies clearly identified an effect of photoperiod on temperature sensitivity for *Fagus sylvatica* (Fu et al. 2019). The fact that day length was not selected in our study might arise from the collinearity with incoming shortwave radiation that was the variable selected by the model instead. Incoming radiation is barely used in phenology studies compared to day length as a proxy for photoperiodism. In addition, studies on photoperiod are based on the temporal sensitivity to light while here we use long-term averages as proxies of biogeographical constraints, which could explain the differences observed. We discussed the fact that incoming radiation should be considered more often in phenological studies (L. 294).

Moreover, some papers used in the review by Way and Montgomery (2014) may have misinterpreted the results. For example, in the Caffara and Donnelly (2011) paper (the reference used to classify *Tilia cordata* as photoperiod sensitive in Way and Montgomery 2014), the researchers compared budbreak under short (8h) and long (16h) photoperiods. As they used temperatures of 22°C for the day and 14°C for the night, longer photoperiod was associated with greater thermal accumulation and, of course, advanced budbreak time (days).

Fu, Y. H., Piao, S., Zhou, X., Geng, X., Hao, F., Vitasse, Y., & Janssens, I. A. (2019). Short photoperiod reduces the temperature sensitivity of leaf-out in saplings of *Fagus sylvatica* but not in horse chestnut. *Global change biology*, 25(5), 1696-1703.

[Comment #R1.15] Line 189-190: Two species names need to be italicised.

[Response #R1.15] We corrected the text font for *A. glutinosa* and *S. aucuparia* L. 183.

Discussion

[Comment #R1.16] Line 217-224: I find this paragraph confusing. As stated, it seems to be making the point that growing season temperature is important for leaf unfolding, without clarifying that it may be an important evolutionary pressure on leaf unfolding (not an important environmental cue for leaf unfolding). It will already be easy for readers to mistake your meaning, given that most studies analyse temporal and not spatial phenology patterns, so you will need to be very clear throughout to avoid confusion.

[Response #R1.16] Thanks for making us notice it. We reformulated L.217-222: “this result indicates that adaptation to long-term mean site biogeographical conditions, including growing season conditions but also a suite of biological interactions that could not be included in this study, constitutes an important evolutionary mechanism to optimize LU at that location, and must be considered in addition to commonly used pre-season temperatures, which do not suffice to explain the spatial distribution of LU”

[Comment #R1.17] Line 236: “may be more important”? The title of this subsection states that radiation IS more important; which is correct?

[Response #R1.17] We corrected “may be more important” by “were more important” L. 269

[Comment #R1.18] Line 242-244: Could you elaborate on these two studies to help support your hypothesis? As currently stated, this hypothesis is rather vague.

[Response #R1.8] We edited L.277-278 and added a reference (Singh et al. 2017) to support our hypothesis: “Increasing evidence also suggests that light modulates internal hormone-regulated growth⁴⁷ and protein production in plants⁴⁸ by affecting signaling pathways of ethylene and abscisic acid, two phytohormones involved in bud set and leaf development⁴⁹ »

Singh, R. K., Svystun, T., Aldahmash, B., Jönsson, A. M., & Bhalerao, R. P. (2017). Photoperiod-and temperature-mediated control of phenology in trees—a molecular perspective. *New Phytologist*, 213(2), 511-524.

[Comment #R1.19] Line 256: I think you mean, “... both SWG and LWG may only be ...”

[Response #R1.19] We corrected “both SWG and LWG can only be” by “both SWG and LWG may be”. L.292

[Comment #R1.20] Line 262: Delete “v”

[Response #R1.20] We deleted “v” L.298.

[Comment #R1.21] Line 275-279: Interesting and reasonable hypothesis! It also provides additional time for rain events that may alleviate drought conditions.

[Response #R1.21] Thanks for this positive feedback. We added a reference to a very recent paper (Gonsamo et al. 2019) which supports our hypothesis L. 312. This paper shows

that earlier leaf unfolding events increased drought during summer based on remote sensing observations.

Gonsamo, A., Ter-Mikaelian, M. T., Chen, J. M., & Chen, J. (2019). Does Earlier and Increased Spring Plant Growth Lead to Reduced Summer Soil Moisture and Plant Growth on Landscapes Typical of Tundra-Taiga Interface?. *Remote Sensing*, 11(17), 1989.

[Comment #R1.22] Line 287: Figure reference here?

[Response #R1.22] We added a reference to Figure 2 L.322.

[Comment #R1.23] Line 288: Do you mean “high drought sensitivity”? Or perhaps “low drought tolerance” would be more clear.

[Response #R1.23] We reformulated L. 323 as « high drought tolerance »

[Comment #R1.24] Line 289: This would be more relevant: “A. glutinosa naturally occurs in wet sites”.

[Response #R1.24] We corrected L. 324 by “A. glutinosa naturally occurs in wet sites”.

[Comment #R1.25] Line 291: Replace “drought tolerance capacity” with “low drought tolerance”.

[Response #R1.25] We replaced “drought tolerance capacity” with “drought tolerance” L.326.

[Comment #R1.26] Line 346-347: Interesting! To drive this point home, could you explain how the GDD definitions would differ? For example, “Northern sites would need to have lower (?) temperature thresholds for GDD than southern sites.”

[Response #R1.26] We moved this paragraph at the beginning of the discussion as suggested by the editor (L.224-257). We edited L. 244-247 as suggested: “Compared to current regional models using constant GDD definition independently of the studied region, our results (Figure 2 & 4) suggested that Northern sites need to have lower temperature threshold for temperature sum and/or lower critical GDD threshold than southern sites. »

We also added a paragraph L. 226-235 to clearly emphasize the relation between LU and GDD and the expected behavior at the regional scale.

Conclusion

[Comment #R1.27] Line 370: Change “control” to “influence”.

[Response #R1.27] Please note that the conclusion has been moved at the end of the introduction as requested by the journal format. We changed “control” by “influence” L. 114.

[Comment #R1.28] Line 382: Delete, “... and always accompanied by analysis of background environmental constraints and evolutionary pressures.” This will not be possible for many studies, due to logistics but also to a lack of knowledge for the specific mechanisms involved in these processes.

[Response #R1.28] We agree with the reviewer’s comment and we deleted the end of the sentence “... and always accompanied by analysis of background environmental constraints and evolutionary pressures.”

[Comment #R1.29] I hope that you find these comments helpful,

[Response #R1.29] Many thanks Dr. Marchin. Yes, your comments were very useful to improve the manuscript.

Renée M. Marchin

Western Sydney University

Reviewer #3 (Remarks to the Author):

[Comment #R3.1] The authors have addressed my concerns and I now recommend acceptance.

[Response #R3.1] We thank the reviewer for the positive feedback on our manuscript.

Reviewer #4 (Remarks to the Author):

I came into this after the revision, asked particularly for a comment on the statistics (collinearity).

[Comment #R4.1] I think the authors treated the issue of collinear predictors very well. The glmnet approach is IMHO one of the best ways to address collinearity for predictive models. I have no issue with this part of the analysis.

[Response #R4.1] We thank the reviewer for this positive feedback on our statistical approach.

[Comment #R4.2] Reading the manuscript, I remain confused about several separate issues, which I detail here.

[Response #R4.2] We agree with the reviewer that the previous version of the manuscript did not do a good job in highlighting the key focus and hypothesis of our study, which could easily lead to a misinterpretation of the results.

In this study we were **not** looking at the temporality of processes **nor** the drivers of LU in spring as commonly assessed in phenological studies, but rather at the biogeographical constraints on LU on the long term and how the long-term environmental conditions constrained spring phenology spatial variability. Thus, we are not looking at the effect of the previous year meteorological conditions on the current year spring phenology. Following the Editor and Reviewer's suggestions we now clearly emphasized the spatial aspect and the underlying hypothesis of our study and we re-edited the introduction and the discussion.

[Comment #R4.3] 1. Line 78 defines LU as function of GDD, while the analysis uses GDD (and other variables) as predictors. I understand the point: process models define LU in this way, and the authors show that this is too simplistic. However, since the authors never give the actual regression "equation" (something like $LU \sim N(\mu = f(\text{GDD5}, \dots), \text{sd}=\text{const.})$ or so), the reader may easily be confused into taking the equation of line 78 as the definition of LU. I would scrap this equation, or at least relegate it out of sight into the supplement.

[Response #R4.3] We followed the reviewer's suggestion and removed the equation of line 78 from the main text to avoid confusion in the definition of LU and moved the equation of GDD to the method section (L. 400).

[Comment #R4.4] Similarly, L85 increases this confusion, as it is not clear whether the authors use LU as defined above, or as independently measured response variable. In the former case, the sentence is strange and should be "LU is a function of GDD and keeping ...". In the latter case, the sentence should be something like "Observed LU might behave differently to the way it is represented in process models, as a function of GDD5."

[Response #R4.4] In the whole study LU comes from field observations and was not modelled. We thus followed the reviewer's suggestion and reformulated L.59 as : "However,

when applied at the regional scale, LU models were not able to accurately reproduce the observed spatial variation of LU¹⁴⁻¹⁷ “to avoid any possible confusion.

We also reformulated the whole paragraph L.54-60 for more clarity: “While the temporal variation in LU and its GDD_{req} is extensively studied, the spatial heterogeneities of LU, and especially of its controls, have been much less studied. Land surface models assume that the drivers of the temporal variation of LU are also determining the spatial gradients in LU, but to our knowledge, this assumption has not been thoroughly tested. The temperature response of LU is often considered constant (spatially and temporally uniform), albeit species-dependent, in phenology models¹⁴. However, when applied at the regional scale, LU models were not able to accurately reproduce the observed spatial variation of LU¹⁴⁻¹⁷. “

[Comment #R4.5] 2. What is actually the RHS of the glmnet model?

Fig. 1 suggests that a range of predictors were used, but Table 1 shows that also their interactions were included. How then did the interaction feature in Fig. 1?

[Response #R4.5] Interactions were not accounted for in this study. Table 1 represents the standard deviation of field observations of LU and GDD, not model outputs from glmnet. We have now edited Table.1 (L.668) caption as well as the method section (L.432-434) to clarify this point.

[Comment #R4.6] Which variable got the partial r^2 of the interaction? Were also non-linear terms (i.e. quadratic) part of the model (or if not, why not: do the authors think that all effects are linear)?

[Response #R4.6] Only single terms were added in the model. Quadratic terms were not a part of the model. We know that environmental effects on phenology are not linear (see the formulation of the GDD model for temperature) when looking at temporal processes, but the current knowledge does not allow to estimate which form should be used for assessing the long-term effects at the spatial scale. As there are already 13 variables, exploring non-linear effects increases the number of variables and the complexity of the interpretation with the risk to dilute the key message. This is the reason why we kept single effects in our analysis and also the reason why we put in parallel results for leaf unfolding and the corresponding GDD.

We edited the method section L.432-434 to clarify this point.

[Comment #R4.7] Was “species” a predictor in the glmnet, and if so, why not as a random effect in interaction with all predictors (as every species’ LU responds differently to the environment, it seems).

[Response #R4.7] We agree with the reviewer that some species show substantial differences to environmental clues. Therefore, we ran the model for each species separately (See Fig.2). Observed effects were globally consistent between species and the few species differences were discussed in the discussion section (L.319). Since we are interested in biogeographical constraints at the regional scale and we observed that species differences are limited regarding the uncertainty in data aggregation, we performed an additional analysis with all species grouped together to see if global behaviors could be detected (Fig. 1 & 2). Moreover, assessing the effect of environment on all species is also consistent with the current representation of phenology in regional studies that group all deciduous trees in a same plant functional type.

We edited the method section L. 451-454 to clarify this point.

[Comment #R4.8] 3. How comes that in Fig. 1 GDD is irrelevant (a and c), but in Fig. 2 it is among the darkest colours for several species, indicating its importance?

[Response #R4.8] In Fig. 1 we only illustrated the results with all data pulled together. Fig. 1a and 1c thus correspond to the last row (ALL) of Fig. 2a and 2b. We edited the caption of Fig. 1 to clarify it (L. 632)

[Comment #R4.9] 4. Fig. 3 a: This figure suggests a tiny effect of aridity on GDD until LU. Is this the basis for claiming that aridity modifies the effect of GDD on LU? If so, I would call the evidence flimsy at best.

[Response #R4.9] Figure 3 illustrates the effect of long-term site aridity on heat requirement. Since αE is an average over 30 years, a value 0.9 already represent sites that face regular drought events. Figure 3 was used to separate drought-prone sites from others (L. 170), the effect of drought pressures on the spatial heterogeneity of LU and associated GDD was assessed with glmnet as we did with all sites (see figure 1 and 2), not from Figure 3. As described L. 196-198, we explored the differences between warm/cold sites and high/low light sites but only a significant effect of drought was observed in the end.

[Comment #R4.10] Fig. 3 b: What's the point of this figure? It seems to be neither referred to in the text, nor interpreted there.

[Response #R4.10] While Fig. 3a showed the distribution of GDD values along the aridity gradient for all species pulled together, Fig. 3b illustrated the constraint of aridity on GDD variability for each species.

We agree that panel a and b were not precisely described in the main text. We thus moved panel a of Figure 3 in Supplementary Figure 4 and now referred only to panel b in the main text (L.172)

[Comment #R4.11] Fig. 4: So, when it is on average warming in winter, then there are fewer chilly days? This blatantly obvious fact warrants a figure? And what is the switch from red to blue supposed to indicate, and where does the turning point of 125 Wm⁻² come from? Again: what's the point of this figure?

[Response #R4.11] We agree that the relationship between chilling and forcing is known for a long time, but this figure did not aim at illustrating this known relationship.

As stated L. 240 of the main text, we looked specifically at this relationship because we observed that GDDreq did not explain well the spatial variability of LU. The current practice in vegetation models or regional studies is to apply the same constant relationship everywhere. Here we show that a large bias is introduced if constant parametrization is used, especially because the local biogeographical constraints are not considered, even when using a chilling-forcing model.

The switch from red to blue represent the gradient in site long-term SWP around the median value (130 W m⁻²). Indeed, we forgot to add the description of the color gradient in the caption. We updated the caption of Figure 4 to describe the color gradient (L.665).

[Comment #R4.12] Overall, I found this MS highly confusing. It seems to make the important point that GDD is not a straightforward predictor of LU, as represented in various physiological models. Fine. The analysis of what actually drives LU could then be made much clearer, without dragging GDD along.

[Response #R4.12] We agree with the reviewer that our key hypothesis and the focus of the study were still confusing. This analysis did not focus on the drivers of LU but on the environmental pressures that explain the observed spatial heterogeneity in LU. Since GDD is often used to describe LU in regional models, we think that keeping in parallel the analysis on LU and GDD, provides an important information to highlight the current misuse of GDD models for regional predictions.

Following the suggestions of the Editor and of Reviewers #1 and #4, we now clearly emphasize the spatial aspect of our study in the introduction (L.54-60; 75-88) and throughout the discussion to avoid any confusion between the temporal and spatial variability of LU. We also rephrased the hypothesis L. 90-99.

[Comment #R4.13] Apparently, TG (mean growing season temperature) seems to be a very important predictor. However, as the other reviewers commented, how can it PREDICT LU, if it comes AFTER LU? The responses to the reviewers seem waffle to me.

[Response #R4.13] Please notice that we are not looking at the temporality of processes nor the drivers of LU in spring but rather at the biogeographical constraints on LU on the long term. Thus, we are not looking at the effect of the previous (or next) year meteorological conditions on the current year spring phenology and we never postulated that growing season temperature can predict LU events. Growing season temperature is a strong biogeographical constraint playing on tree development (see Discussion L. 212-222 and L.343-367). Our analysis highlights that long-term environmental pressures on tree phenology do not limit to spring meteorological conditions and that TG is a better proxy of the spatial heterogeneity in LU than GDD. In addition, this result is in line with recent “temporal” studies which highlighted an indirect effect of growing season temperatures on next year flushing events by altering bud set and bud development (See. Discussion section L. 355-357).

As suggested by the Editor and Reviewer #1 we have now edited the introduction and the discussion section to clearly emphasize the main hypothesis of our study regarding the impact of long-term biogeographical constraints (evolutionary pressures) on leaf unfolding that could explain the observed spatial heterogeneity in its response and the fact that we are not looking here at the environmental drivers of the inter-annual variability in LU.